# MULTI-SCALE CONDITIONAL GENERATIVE MODELING FOR MICROSCOPIC IMAGE RESTORATION

## ABSTRACT

The advance of diffusion-based generative models in recent years has revolution-ized state-of-the-art (SOTA) techniques in a wide variety of image analysis and synthesis tasks, whereas their adaptation on image restoration, particularly within computational microscopy remains theoretically and empirically underexplored. In this research, we introduce a multi-scale generative model that enhances con-ditional image restoration through a novel exploitation of the Brownian Bridge process within wavelet domain. By initiating the Brownian Bridge diffusion process specifically at the lowest-frequency subband and applying generative adversarial networks at subsequent multi-scale high-frequency subbands in the wavelet do-main, our method provides significant acceleration during training and sampling while sustaining a high image generation quality and diversity on par with SOTA diffusion models. Experimental results on various computational microscopy and imaging tasks confirm our method's robust performance and its considerable reduc-tion in its sampling steps and time. This pioneering technique offers an efficient image restoration framework that harmonizes efficiency with quality, signifying a major stride in incorporating cutting-edge generative models into computational microscopy workflows.

## 1 INTRODUCTION

Within the last decade, the landscape of image synthesis has been radically transformed by the advent of generative models (GMs) (Song et al., 2020b; Ho et al., 2020; Song & Ermon, 2019). Among their broad success in various image synthesis applications, image restoration, including super-resolution, shadow removal, inpainting, etc, have caught much attention due to their importance in various practical scenarios. Image restoration aims to recover high-quality target image from low-quality images measured by an imaging system with assorted degradation effects, e.g., downsampling, aberration and noise. Numerous tasks in photography, sensing and microscopy can be formulated as image restoration problems, and therefore the importance of image restoration algorithms is self-evident in practical scenarios (Isola et al., 2017; Kupyn et al., 2018; Weigert et al., 2018; Wang et al., 2019).

Due to the ill-posedness of most image restoration problems, the application of generative learning becomes crucial for achieving high-quality image reconstruction. The wide applications of deep learning (DL)-based generative models in image restoration began with the success of generative adversarial networks (GANs) (Goodfellow et al., 2014). The emergence of conditional adversarial learning achieved unprecedented success and outperformed classical algorithms in various applica-tions of computational imaging and microscopy (Zhu et al., 2017; Isola et al., 2017; Wang et al., 2018c; Kupyn et al., 2018; Nazeri et al., 2018; Karras et al., 2019; Weigert et al., 2018; Wang et al., 2019). Although GANs have achieved remarkable success in image restoration, they are also known to be prone to training instability and mode collapse. These issues significantly restrict the diversity of outputs produced by GAN models (Kodali et al., 2017; Gui et al., 2021). Recently, diffusion models (DMs) were intensively studied and outperformed GANs in various image generation tasks (Ho et al., 2020; Dhariwal & Nichol, 2021). Derived from the stochastic diffusion process, DMs employ neural networks to approximate the reverse diffusion process and sample target images from white noise with high quality and good mode coverage. Moreover, DMs have also been applied to image restoration, including super-resolution, dehazing, colorization, etc (Saharia et al., 2022b; Rombach et al., 2022; Li et al., 2023; Batzolis et al., 2021; Saharia et al., 2022a; Luo et al., 2023a). Most of the

existing methods regard the low-quality image as one additional condition in the reverse process and utilize it as one of the network arguments for inference. However, these image restoration models lack a clear modelling of the conditional image in the forward process and a theoretical guarantee that the terminal state of the diffusion is closely related to and dependent on the condition.

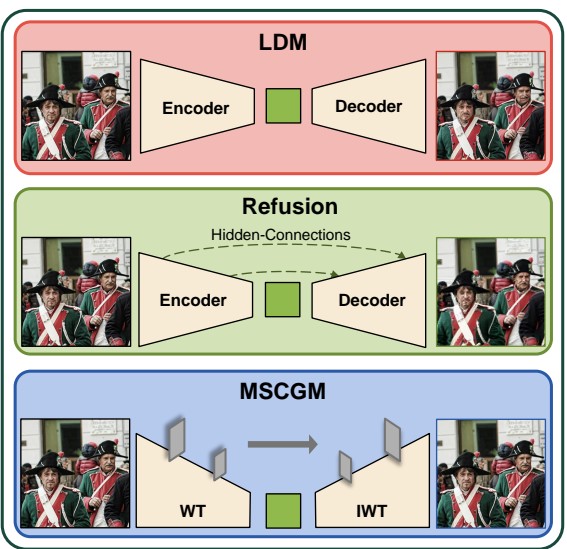

Figure 1: **Inherent information loss in autoencoder backbones.** The same high-quality image passes the three generative models without going through the diffusion process. Public pre-trained checkpoints are used for LDM (Rombach et al., 2022) and Refusion (Luo et al., 2023b). LDM and Refusion suffer from reconstruction loss due to lossy autoencoder backbones while the wavelet and inverse wavelet transform pairs are lossless.

Besides, compared to GANs, DMs are further limited by time-consuming sampling and iterative refinement. Many efforts have been made to improve the sampling speed while ensuring the quality of image generation, among which the Latent Diffusion Model (LDM) (Rombach et al., 2022) stands out as a notable example. LDM projects the diffusion process into a low-dimensional latent space of a pre-trained autoencoder, and hence greatly reduces the computational resource required for high-resolution image generation and restoration. The success of LDM inspired a few studies leveraging the latent representations of pretrained networks to reconstruct high-quality images from low-quality images (Yin et al., 2022; Luo et al., 2023a). However, the performance of diffusion models in these approaches is highly dependent on the quality of latent representations of their pretrained backbones. There exist inevitable trade-offs between the compactness of the latent space and the reconstruction fidelity of the autoencoder, and between the normality of its latent distribution and its sample diversity (Saharia et al., 2022b; Li et al., 2022; Luo et al., 2023a). Some study, e.g., Refusion, has explored to improve the fidelity of autoencoder by adding hidden connections between the encoder and decoder (Luo et al., 2023b), whereas the inherent information loss of learned compression cannot be completely eliminated. Though these methods reported improved perceptual scores between their outputs and reference images, in image restoration tasks where pixel-wise structural correspondence is emphasized, the reconstruction quality is constrained by the fidelity of autoencoder backbones, producing noticeable distortions and hallucinations as shown in Figure 1.

On the other hand, microscopy image restoration problems are usually composed of complicated degradation models, sparse image distributions and strict constraints on structural correspondence of the outputs to the conditional images (Meiniel et al., 2018; Solomon et al., 2018; Zhao et al., 2022). To preserve the signal sparsity of common biological samples and local feature correspondence to the conditional image, existing methods leveraged CNN-based models and trained on carefully registered datasets (Rivenson et al., 2017; Wang et al., 2019; Wu et al., 2019). Moreover, to enhance training and inference stability, conditional GAN were widely utilized to directly predict the corresponding ground truth image in a deterministic manner (Wang et al., 2019). However, the adaptability and effectiveness of advanced generative models like DMs to microscopy image restoration tasks are under-explored.

To address the aforementioned limitations, here we introduce a novel multi-scale conditional generative model (MSCGM) for image restoration based on multi-scale wavelet transform and Brownian bridge stochastic process. For one thing, multi-scale wavelet transform effectively and losslessly compresses the spatial dimensions of conditional images, eliminating the lossy encoding process of current autoencoder-based DMs. Notably, our method does not involve the pre-training of autoencoders in competitive methods on a sufficiently large and diverse dataset sampled from the image domain. For another, Brownian bridge stochastic process incorporates the modelling of low-

quality conditional image into both the forward and reverse diffusion process and better utilizes the information of the conditional images. Besides, we theoretically analyze the distributions of low- and high-frequency wavelet subbands and apply Brownian bridge diffusion process (BBDP) and adversarial learning to the multi-scale generation of low- and high-frequency subbands, respectively. In sum, the contributions of this work are three-fold:

1. We are the first to factorize the conditional image generation within multi-scale wavelet domains, establishing a theoretical groundwork for a multi-scale conditional generative modeling;

2. Capitalizing on the unique distribution characteristics of wavelet subbands, we propose the innovative MSCGM, which seamlessly integrates BBDP and adversarial learning;

3. We evaluate the MSCGM on various image restoration tasks, demonstrating its superior performance in both sampling speed and image quality than competitive methods.

## 2 RELATED WORKS

### 2.1 MICROSCOPY IMAGE RESTORATION

As an important branch of computational imaging, computational microscopy springs up in recent years and aims to restore high-quality, multi-dimensional images from low-quality, low-dimensional measurements, usually with under-resourced equipment. Since the first work on microscopy image super-resolution reported in 2017 Rivenson et al. (2017), DL has enabled a wide spectrum of novel applications that were impossible with conventional optical technologies, e.g., microscopy image super-resolution surpassing the physical resolution limit of microscopic imaging systems Wang et al. (2019), volumetric imaging reconstructing 3D sample volumes from sparse 2D measurements Wu et al. (2019), and virtual image labelling to match the contrast conventionally provided by chemical or biological markers Rivenson et al. (2019). Compared to general image restoration in computational imaging, microscopy image restoration mainly differs in two aspects: (1) The degeneration process, including the transfer function, noise and aberration of the imaging system is generally complex, unknown and hard to measure precisely; and such degeneration process could vary significantly in real-world scenarios due to the variations of subjects, hardware and imaging protocols. (2) Strict pixel-wise correspondence between output and ground truth images and consistency with physical laws are generally emphasized Barbastathis et al. (2019).

### 2.2 GENERATIVE MODELS

GANs are well-known for generating high-qaulity, photorealistic samples rapidly Goodfellow et al. (2014); Gui et al. (2021). Through training a discriminator that tells ground truth images apart from fake images generated by the generator network, GANs outperformed traditional CNNs trained with hand-crafted, pixel-based structural losses such as $L_1$ and $L_2$ by providing a high-level, learnable perceptual objective. Conditional GANs such as Pix2Pix Isola et al. (2017), pix2pix HD Wang et al. (2018c) and starGAN Choi et al. (2018) have been successfully applied in a wide spectrum of image-to-image translation and image restoration tasks, including image colorization Nazeri et al. (2018), style transfer Karras et al. (2019), image deblurring Kupyn et al. (2018), etc. Unsupervised image-to-image translation has also been extensively explored, such as cycleGAN Zhu et al. (2017), UNIT Liu et al. (2017), DualGAN Yi et al. (2017), etc. In the fields of biomedical and microscopy imaging, researchers have also explored the applications of GANs, e.g., reconstructing low-dose CT and MRI images Yang et al. (2018); Hammernik et al. (2018), denoising microscopy images Weigert et al. (2018), super-resolving diffraction-limited microscopy images Wang et al. (2019), among others. Recently, WGSR Korkmaz et al. (2024) has also provided wavelet guided GAN model for image super-resolution tasks.

Alternatively, transformer and related architectures have recently emerged and shown superior performance over convolutional neural networks (CNNs)-based GANs. Swin transformer Liu et al. (2021) and SwinIR Liang et al. (2021) have established a strong transformer-based baseline with competitive performance to CNNs. TransGAN substituted CNNs in the common GAN framework with transformers and improved the overall performance Jiang et al. (2021). More recently, diffusion models (DMs) have been introduced and proved to be the state-of-the-art generative model in various image generation benchmarks Ho et al. (2020); Nichol & Dhariwal (2021). Dhariwal et

al. reported diffusion models beat GANs on various image synthesis tasks Dhariwal & Nichol (2021). Nevertheless, the democratization of diffusion models was limited by its huge demand for computational resources in both training and sampling. Recent advancements in fast sampling diffusion models Xiao et al. (2021); Rombach et al. (2022); Song et al. (2020a); Phung et al. (2023); Kong & Ping (2021); Ho et al. (2022) have accelerated the sampling process in image generation tasks. DDGAN Xiao et al. (2021) effectively combines the strengths of GANs with diffusion principles for quicker outputs, while LDM utilizes a compressed latent space to speed up generation. DDIM Song et al. (2020a) optimizes denoising steps for efficiency, FastDPM Kong & Ping (2021) focuses on algorithmic improvements for rapid sampling, and CDM Ho et al. (2022) employs a staged approach to simplify the diffusion process. Furthermore, recent works like **?**

## 3 METHODS

### 3.1 PRELIMINARIES

The outstanding success and wide applications of score-based diffusion models have been witnessed in the past years. Generally, for a Gaussian process $\{\boldsymbol{x}_t, t = 1, \cdots, T\}$ defined as:

$$q(\boldsymbol{x}_t|\boldsymbol{x}_{t-1}) = N(\boldsymbol{x}_t; \sqrt{1 - \beta_t}\boldsymbol{x}_{t-1}, \beta_t \boldsymbol{I}), t = 1, \cdots, T, \tag{1}$$

the denoising DM attempts to solve the reverse process by parameterizing the conditional reverse distribution as:

$$p_\theta(\boldsymbol{x}_{t-1}|\boldsymbol{x}_t) = N\Big(\boldsymbol{x}_t; \frac{1}{\sqrt{\alpha_t}}\big(\boldsymbol{x}_t - \frac{1 - \alpha_t}{\sqrt{1 - \bar{\alpha}_t}}\boldsymbol{\epsilon}_\theta(\boldsymbol{x}_t, t)\big), \sigma_t \boldsymbol{I}\Big). \tag{2}$$

Here $\alpha_t, \bar{\alpha}_t, \beta_t, \sigma_t$ are constants, and $\boldsymbol{\epsilon}_\theta$ is the network estimating the mean value of the reverse process. Despite the high image quality achieved by the denoising process, the total sampling steps $T$ can be very large and make the sampling process time-consuming.

**Theorem 1.** *For a given error $\varepsilon$ between the generated distribution $p_\theta$ and the true distribution $p$, the sampling steps needed can be expressed by Guth et al. (2022):*

$$T = O(\varepsilon^{-2}\kappa^3), \tag{3}$$

*where $\kappa$ is the condition number of the covariance matrix of $p$.*

The proof of Theorem 1 is detailed in Appendix B.

Therefore, for highly non-Gaussian distributed images, e.g., microscopy images (please refer to Appendix C, where we theoretically and statistically demonstrate the high degree of non-Gaussianity of microscopy datasets), which tends to have high sparsity, resolution and contrast, standard diffusion models may not be practical due to their slow sampling speed and large sampling steps required for such distributions.

To overcome this limitation, we turned to multi-scale wavelet transform (as detailed in Appendix D), which offers an excellent latent space (i.e., wavelet domain) for generative modeling. The wavelet domain not only facilitates lossless compression but also provides low-frequency coefficients with near-Gaussian distributions. Consequently, we transform the diffusion process into the wavelet domain, thereby introducing a novel multi-scale wavelet-based generative model. It can be shown that the diffusion in wavelet domain is a dual problem to the original diffusion in the spatial domain. For a detailed explanation of this duality, please see Appendix E. Due to the distinct characteristics of wavelet coefficients in low- and high-frequency subbands (as detailed in Appendix C), we adopt different generative modeling approaches for the low- and high-frequency coefficients, marking a key innovation in our work. Specifically, while the low-frequency wavelet coefficients exhibit a Gaussian tendency, the high-frequency coefficients are sparse and non-Gaussian. Therefore, for the low-frequency coefficients, we employed the Brownian Bridge Diffusion Process (BBDP) Li et al. (2023), and for the high-frequency coefficients, we utilized a Generative Adversarial Network (GAN) based generative method.

### 3.2 BROWNIAN BRIDGE DIFFUSION PROCESS

We leverage BBDP to better model the conditional diffusion process and apply it to image restoration. Image restoration tasks focus on the generation of the target image $\boldsymbol{x}_0 \in \mathbb{R}^{H \times W \times C}$ from a conditional

image $\boldsymbol{y} \in \mathbb{R}^{H \times W \times C}$. Most existing DMs tackle the conditional image $\boldsymbol{y}$ as an additional input argument to $\boldsymbol{\epsilon}_\theta$, without integrating the conditional probability distribution on $\boldsymbol{y}$ into the diffusion theory. Different from the standard diffusion process, we adapt the Brownian bridge process and derive a conditional diffusion process for image-to-image translation, termed as BBDP.

**Definition 2** (Forward Brownian bridge process). The forward Brownian bridge with initial state $\boldsymbol{x}_0$ and terminal state $\boldsymbol{y}$ is defined as

$$q(\boldsymbol{x}_t|\boldsymbol{x}_0, \boldsymbol{x}_T = \boldsymbol{y}) = N\left((1 - \frac{t}{T})\boldsymbol{x}_0 + \frac{t}{T}\boldsymbol{y}, \frac{t(T-t)}{T^2}\boldsymbol{I}\right), \tag{4}$$

By denoting $m_t = \frac{t}{T}$ and $\delta_t = \frac{t(T-t)}{T^2}$, we can reparameterize the distribution of $\boldsymbol{x}_t$ as:

$$\boldsymbol{x}_t = \boldsymbol{x}_0 + m_t(\boldsymbol{y} - \boldsymbol{x}_0) + \sqrt{\delta_t}\boldsymbol{\epsilon}_t, \boldsymbol{\epsilon}_t \sim N(\boldsymbol{0}, \boldsymbol{I}) \tag{5}$$

Following existing works on diffusion models, a network is trained to estimate $\boldsymbol{x}_0$ from $\boldsymbol{x}_t, \boldsymbol{y}$; in other words, a network $\boldsymbol{\epsilon}_\theta$ is trained to estimate $m_t(\boldsymbol{y} - \boldsymbol{x}_0) + \sqrt{\delta_t}\boldsymbol{\epsilon}_t$.

The loss function of $\boldsymbol{\epsilon}_\theta$ is defined as below where $\gamma_t$ is the weight for each $t$:

$$L = \Sigma_t \gamma_t \mathbb{E}_{(\boldsymbol{x}_0, \boldsymbol{y}), \boldsymbol{\epsilon}_t} \|m_t(\boldsymbol{y} - \boldsymbol{x}_0) + \sqrt{\delta_t}\boldsymbol{\epsilon}_t - \boldsymbol{\epsilon}_\theta(\boldsymbol{x}_t, \boldsymbol{y}, t)\|_2^2, \tag{6}$$

**Theorem 3.** *The reverse process can be shown to be a Gaussian process with mean $\boldsymbol{\mu}_t'(\boldsymbol{x}_t, \boldsymbol{x}_0, \boldsymbol{y})$ and variance $\delta_t'\boldsymbol{I}$*

$$p(\boldsymbol{x}_{t-1}|\boldsymbol{x}_t, \boldsymbol{x}_0, \boldsymbol{y}) = N(\boldsymbol{x}_{t-1}; \boldsymbol{\mu}_t'(\boldsymbol{x}_t, \boldsymbol{x}_0, \boldsymbol{y}), \delta_t'\boldsymbol{I}), \tag{7}$$

$$\boldsymbol{\mu}_t'(\boldsymbol{x}_t, \boldsymbol{x}_0, \boldsymbol{y}) = c_{xt}\boldsymbol{x}_t + c_{yt}\boldsymbol{y} - c_{\epsilon t}\boldsymbol{\epsilon}_\theta(\boldsymbol{x}_t, \boldsymbol{y}, t) \tag{8}$$

$$\delta_t' = \frac{\delta_{t|t-1}\delta_{t-1}}{\delta_t}. \tag{9}$$

Equation 8 indicates the posterior sampling process of BBDP and the training process of BBDP is detailed here. The input is the low-frequency wavelet coefficients of the original image and the output is the low-frequency wavelet coefficients of the restored image.

---

**Algorithm 1** Training of BBDP

---

1: **repeat**
2:    $\boldsymbol{x}_0, \boldsymbol{y} \sim q(\boldsymbol{x}_0, \boldsymbol{y})$
3:    $t \sim \text{Uniform}([1, \cdots, T])$
4:    $\boldsymbol{\epsilon}_t \sim N(\boldsymbol{0}, \boldsymbol{I})$
5:    Take gradient descent step on
     $\nabla_\theta \|m_t(\boldsymbol{y} - \boldsymbol{x}_0) + \sqrt{\delta_t}\boldsymbol{\epsilon}_t - \boldsymbol{\epsilon}_\theta(x_t, y, t)\|_2^2$
6: **until** converged

---

The proof of the expression of $c_{xt}, c_{yt}, c_{\epsilon t}$ is provided as below:

*Proof of Theorem 3.* Brownian bridge process can prove to be Markovian and hence defined alternatively by its one-step forward process. The one-step forward process $q(\boldsymbol{x}_t|\boldsymbol{x}_{t-1}, \boldsymbol{y})$ can be derived as:

$$q(\boldsymbol{x}_t|\boldsymbol{x}_{t-1}, \boldsymbol{y}) = N(\frac{1 - m_t}{1 - m_{t-1}}\boldsymbol{x}_{t-1} + (m_t - \frac{1 - m_t}{1 - m_{t-1}}m_{t-1})\boldsymbol{y}, \delta_{t|t-1}\boldsymbol{I}). \tag{10}$$

Here $\delta_{t|t-1} = \delta_t - \delta_{t-1}\frac{(1-m_t)^2}{(1-m_{t-1})^2}$ and $m_t = \frac{t}{T}$.

The reverse process can be derived by Bayesian formula and shown to be a Gaussian process with mean $\mu_t'(x_t, x_0, y)$ and variance $\delta_t'\boldsymbol{I}$:

$$p(\boldsymbol{x}_{t-1}|\boldsymbol{x}_t, \boldsymbol{x}_0, \boldsymbol{y}) = \frac{p(\boldsymbol{x}_t|\boldsymbol{x}_{t-1}, \boldsymbol{y})p(\boldsymbol{x}_{t-1}|\boldsymbol{x}_0, \boldsymbol{y})}{p(\boldsymbol{x}_t|\boldsymbol{x}_0, \boldsymbol{y})} = N(\boldsymbol{x}_{t-1}; \boldsymbol{\mu}_t'(x_t, x_0, y), \delta_t'\boldsymbol{I}), \tag{11}$$

$$\boldsymbol{\mu}_t'(\boldsymbol{x}_t, \boldsymbol{x}_0, \boldsymbol{y}) = \frac{\delta_{t-1}}{\delta_t}\frac{1 - m_t}{1 - m_{t-1}}\boldsymbol{x}_t + (1 - m_{t-1})\frac{\delta_{t|t-1}}{\delta_t}\boldsymbol{x}_0 + (m_{t-1} - m_t\frac{\delta_{t-1}}{\delta_t}\frac{1 - m_t}{1 - m_{t-1}})\boldsymbol{y}, \tag{12}$$

$$\delta_t' = \frac{\delta_{t|t-1}\delta_{t-1}}{\delta_t}. \tag{13}$$

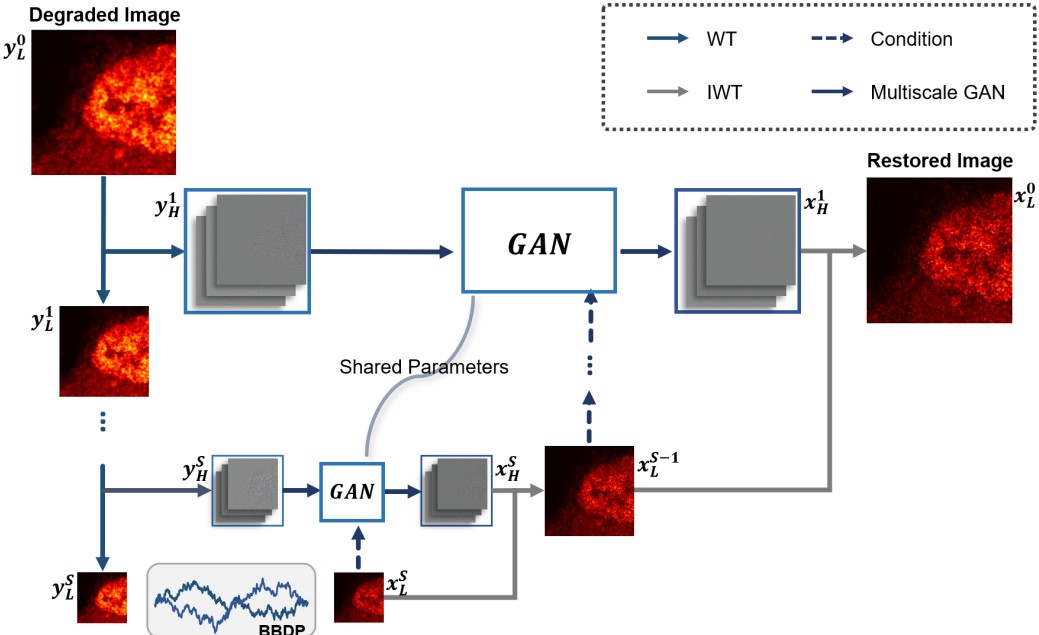

Figure 2: **Schematic diagram of MSCGM.** The conditional image $y_L^0$ is first decomposed by multi-scale wavelet transform (WT). In the coarsest wavelet layer, a BBDP transforms the low-frequency subband of conditional image to the low-frequency subband of the target image. A multi-scale adversarial learning process transforms subsequent high-frequency subbands of conditional image to the high-frequency subbands of the target image and recovers the full-resolution image using inverse wavelet transform (IWT). $y_L^i$ and $y_H^i$ represent the low- and high-frequency wavelet coefficients of the conditional image at the $i^{th}$ level of wavelet transform, respectively. Similarly, $x_L^i$ and $x_H^i$ denote the low-and high-frequency wavelet coefficients of the target image at the $i^{th}$ level of wavelet transform.

Utilizing the estimate of $\boldsymbol{x}_0$ by $\epsilon_\theta$ in Eq. 5, we could eliminate $\boldsymbol{x}_0$ in Eq. 12 and rewrite Eq. 12 as Li et al. (2023):

$$\boldsymbol{\mu}_t'(\boldsymbol{x}_t, \boldsymbol{y}) = \frac{\delta_{t-1}}{\delta_t}\frac{1-m_t}{1-m_{t-1}}\boldsymbol{x}_t + (m_{t-1} - m_t\frac{\delta_{t-1}}{\delta_t}\frac{1-m_t}{1-m_{t-1}})\boldsymbol{y}$$
$$+ (1 - m_{t-1})\frac{\delta_{t|t-1}}{\delta_t}(\boldsymbol{x}_t - \boldsymbol{\epsilon}_\theta(\boldsymbol{x}_t, \boldsymbol{y}, t)) = c_{xt}\boldsymbol{x}_t + c_{yt}\boldsymbol{y} - c_{\epsilon t}\boldsymbol{\epsilon}_\theta(\boldsymbol{x}_t, \boldsymbol{y}, t), \tag{14}$$

$$c_{xt} = \frac{\delta_{t-1}}{\delta_t}\frac{1-m_t}{1-m_{t-1}} + (1-m_{t-1})\frac{\delta_{t|t-1}}{\delta_t}$$
$$c_{yt} = m_{t-1} - m_t\frac{\delta_{t-1}}{\delta_t}\frac{1-m_t}{1-m_{t-1}} \tag{15}$$
$$c_{\epsilon t} = (1-m_{t-1})\frac{\delta_{t|t-1}}{\delta_t}.$$

### 3.3 MULTI-SCALE CONDITIONAL GENERATIVE MODEL

Wavelet transform, with its theoretical details outlined in Appendix D, is characterized by an orthogonal transform matrix $\boldsymbol{A} \in \mathbb{R}^{N^2 \times N^2}$. The wavelet transform decomposes an image $\boldsymbol{x} \in \mathbb{R}^{N^2}$ to one low-frequency (LL) subband $\boldsymbol{x}_L^1 \in \mathbb{R}^{\frac{N^2}{4}}$ and remaining high-frequency subbands $\boldsymbol{x}_H^1 \in \mathbb{R}^{\frac{3N^2}{4}}$.

**Definition 4** (Multi-scale wavelet decomposition of conditional image generation). With multi-scale wavelet transformation, we can reformulate the conditional probability distribution of $\boldsymbol{x}_0$ on $\boldsymbol{y}$ as

$$p(\boldsymbol{x}_0|\boldsymbol{y}) = \Pi_{k=1}^S p(\boldsymbol{x}_H^k|\boldsymbol{x}_L^k, \boldsymbol{y}_H^k)p(\boldsymbol{x}_L^1|\boldsymbol{y}_L^S), \tag{16}$$

where $S$ denotes the maximum scale and

$$(\boldsymbol{x}_H^1, \boldsymbol{x}_L^1)^T = A\boldsymbol{x}_0, \ (\boldsymbol{x}_H^{k+1}, \boldsymbol{x}_L^{k+1})^T = A\boldsymbol{x}_L^k, \ k = 1, \dots \tag{17}$$

Different from existing approaches, our method leverages *BBDP* and *adversarial learning* process inspired by GANs to handle low- and high-frequency subbands at various scales respectively, and the schematic diagram of our model is illustrated in Fig. 2. For the coarsest level low-frequency subband $\boldsymbol{x}_L^S$, due to the whitening effect of the low-frequency subband after wavelet transform, DMs can effectively and efficiently approximate $p(\boldsymbol{x}_L^S|\boldsymbol{y}_L^S)$ with fewer sampling steps, while generating diverse and photorealistic images. For another, though the conditional distribution of high-frequency subbands deviates from unimodal Gaussian distributions considerably, the multi-scale adversarial learning process is able to approximate their multi-modal distribution and sample the full-resolution images rapidly in a coarse-to-fine style. Since the BBDP at the coarsest level produces samples with good diversity and fidelity, the possibility of mode collapse commonly observed in pure GAN models can be minimalized.

We adopt the Wasserstein distances between fake and real images Arjovsky et al. (2017) to optimize the generator $G$ and discriminator $D$. We adopted a pixel-wise L2 loss and a structural similarity index loss to penalize local and global mismatch, respectively. The training loss for multi-scale adversarial learning process is:

$$L_G = \sum_{k=1}^{S} \left[ \lambda(G(\boldsymbol{x}_L^k, \boldsymbol{z}^k) - \boldsymbol{x}_H^k)^2 + \nu(1 - \text{SSIM}(G(\boldsymbol{x}_L^k, \boldsymbol{z}^k), \boldsymbol{x}_H^k)) - \alpha D(G(\boldsymbol{x}_L^k, \boldsymbol{z}^k)) \right] \quad (18)$$

$$L_D = \sum_{k=1}^{S} \left( D(G(\boldsymbol{x}_L^k, \boldsymbol{z}^k)) - D(\boldsymbol{x}_H^k) \right) \quad (19)$$

Here $\boldsymbol{z}^k$ refers to random white noise at scale $k$, $\text{SSIM}(\cdot, \cdot)$ is the structural similarity index measure Wang et al. (2004). The complete sampling process of our model is detailed here:

---
**Algorithm 2** Sampling

---
1: Sample $\boldsymbol{y} \sim q(\boldsymbol{y})$
2: Wavelet transform $S$ times to get $\{\boldsymbol{y}_L^S, \boldsymbol{y}_H^S, \cdots, \boldsymbol{y}_H^1\}$
3: **for** $t = T, T-1, \cdots, 1$ **do**
4:     $\boldsymbol{z} \sim N(\boldsymbol{0}, \boldsymbol{I})$ if $t > 1$, else $\boldsymbol{z} = \boldsymbol{0}$
5:     $\boldsymbol{x}_{t-1,L}^S = c_{xt}\boldsymbol{x}_{t,L}^S + c_{yt}\boldsymbol{y}_L^S - c_{\epsilon t}\boldsymbol{\epsilon}_\theta(\boldsymbol{x}_{t,L}^S, \boldsymbol{y}_L^S, t) + \sqrt{\delta_t'}\boldsymbol{z}$
6: **end for**
7: $\boldsymbol{x}_L^S = \boldsymbol{x}_{0,L}^S$
8: **for** $k = S, S-1, \cdots, 1$ **do**
9:     $\boldsymbol{x}_H^k = G(\boldsymbol{x}_L^k, \boldsymbol{y}_H^k, z^k)$
10:     $\boldsymbol{x}_L^{k-1} = A^T(\boldsymbol{x}_H^k, \boldsymbol{x}_L^k)^T$
11: **end for**
12: Return $\boldsymbol{x}_0 = \boldsymbol{x}_L^0$

---

## 4 EXPERIMENTS

In this section, we first elucidate the design and training details of our method, as well as the preparation of training and testing datasets. Then, we evaluate our method on various image restoration tasks in computational and microscopy imaging, and compare it with baseline methods.

### 4.1 EXPERIMENTAL SETUP AND IMPLEMENTATION DETAILS

For the BBDP at the coarsest wavelet scale, we adapt the UNet architecture with multi-head attention layers as practiced in Nichol & Dhariwal (2021). The number of sampling steps is set as 1000 for training. The Brownian bridge diffusion model (BBDM) baseline Li et al. (2023) is implemented at the full resolution scale without wavelet decomposition, and the same 1000 discretization step is used for training. The training of IR-SDE (image restoration stochastic differential equation) Luo et al. (2023a) and Refusion Luo et al. (2023a) follow the training setups as their original setups.

Inspired by Chen et al. (2022), the generator adopt a similar architecture to NAFNet. The generator contains 36 NAFBlocks distributed at 4 scales. A $2 \times 2$ convolutional layer with stride 2 doubling the

| Methods | Trainable Params.↓ | Sampling Time (s)↓ | PSNR (dB)↑ | | | SSIM↑ | | |
|---|---|---|---|---|---|---|---|---|
| | | | DIV2K | Set5 | Set14 | DIV2K | Set5 | Set14 |
| IR-SDE | 135.3M | 19.51 | 23.54 | 26.73 | 22.71 | 0.56 | 0.72 | 0.54 |
| ReFusion | 131.4M | 17.25 | 21.39 | 22.82 | 22.13 | 0.43 | 0.52 | 0.49 |
| BBDM | 124.7M | 32.29 | 31.50 | 31.60 | 30.39 | 0.66 | 0.76 | 0.68 |
| MSCGM | 192.5M | **2.28** | **31.66** | **32.33** | **30.79** | **0.72** | **0.85** | **0.71** |

Table 1: **Comparison of IR-SDE, ReFusion, BBDM and our method (MSCGM) on $4\times$ super resolution experiment.** Sample steps are set as 1000 for all methods. Metrics are calculated on $256 \times 256$ center-cropped patches of DIV2K validation set, Set 5 and Set 14. Entries in bold indicate the best performance achieved among the compared methods. Sampling time is measured at the same resource cost.

channels connects adjacent scales in the encoding (downsampling) path, and a $1 \times 1$ convolutional layer with pixel shuffle layer connects adjacent scales in the decoding (upsampling)superre path. The number of channels in the first NAFBlock is 64. The discriminator consists of 5 convolutional blocks and 2 dense layers at the end, and each convolutional block halves the spatial dimension but doubles the number of channels. The number of channels for the first convolutional block in the discriminator is 64. The details about training and dataset are elucidated in Appendix F.

## 4.2 EVALUATION METRICS

Peak Signal-to-Noise Ratio (PSNR) is commonly used to measure the quality of reconstruction in generated images, with higher values indicating better image quality. Structural Similarity Index Measure (SSIM) Wang et al. (2004) assesses the high-level quality of images by focusing on changes in structural information, luminance, and contrast. Fréchet Inception Distance (FID) score Heusel et al. (2017) is used in generative models like GANs to compare the distribution of generated images against real ones, where lower FID values imply images more similar to real ones, indicating higher quality.

## 4.3 RESULTS AND COMPARISON

In this section, we first evaluate and compare our method with the current state-of-the-art model (cross-modality super-resolution (CMSR)Wang et al. (2019)) on two microscopy image restoration tasks with different samples, and the experimental results demonstrate that our method achieves the **best performance** on these tasks. To further validate the model's generalization capability on standard nature image datasets, we then conduct comparative evaluations on three different natural image restoration tasks against multiple baselines, achieving similarly excellent restoration results.

First, we apply our method to microscopy images, where the degradation process is complex and unknown. Given the pronounced contrast and sparsity inherent in microscopy images, it is crucial to use generative models capable of handling multi-modal distributions to adapt effectively to complex microscopy datasets. We utilize our method to perform super-resolution on microscopy images of nano-beads and HeLa cells, transforming diffraction-limited confocal images to achieve resolution beyond the optical diffraction limit and match the image quality of STED microscopy. Next, we assess the adaptability of our method to various image restorations tasks of natural images, including a $4\times$ super-resolution task on DIV2K dataset, a shadow removal task on natural images (ISTD dataset) and on a low-light image enhancement task on natural images (LOL dataset). Through comparison against competitive methods on various testbeds, we demonstrate the superior effectiveness and versatility of our method for image restoration.

### 4.3.1 MICROSCOPY IMAGE RESTORATION

**Microscopy Image Super-resolution:** We evaluate our method on microscopy image super-resolution tasks and compare it with existing generative models in this field. Unlike natural image super-resolution, the LR images are not downsampled but sampled at the same spatial frequency as the HR images. However, the LR images are limited by the optical diffraction limit, which is equivalent to a convolution operation on the HR images with a low-pass point spread function (PSF). We apply our method to confocal (LR) images of fluorescence nanobeads to evaluate its capability to overcome the

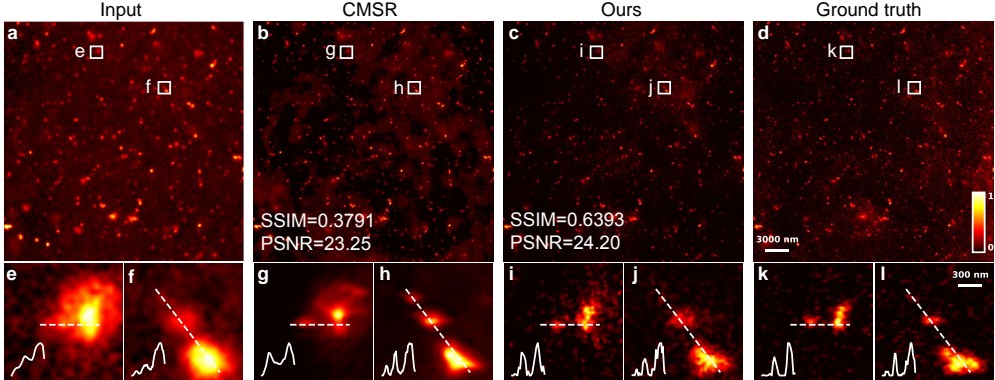

Figure 3: **Comparison of CMSR Wang et al. (2019) and our method on microscopy image super-resolution of nanobeads.** (a) Input images captured by a confocal microscope, (b, c) SR outputs of CMSR and our method, and (d) ground truths captured by an STED microscope of the same FOV. (e-l) Zoom-in regions marked by the corresponding white boxes in (a-c). Cross-section intensity values along the dashed line are plotted.

optical diffraction limit (see Methods for sample and dataset details). Figure 3 illustrates one typical field-of-view (FOV) of nanobead samples ($\sim$ 20-nm) captured using confocal and STED microscopy. The dimensions of nanobeads are considerably smaller than the optical diffraction limit ($\sim$ 250-nm) and nearby beads cannot be distinguished in confocal images. MSCGM and CMSR model are trained on the same training data and learn to transform LR confocal images to match HR STED images.

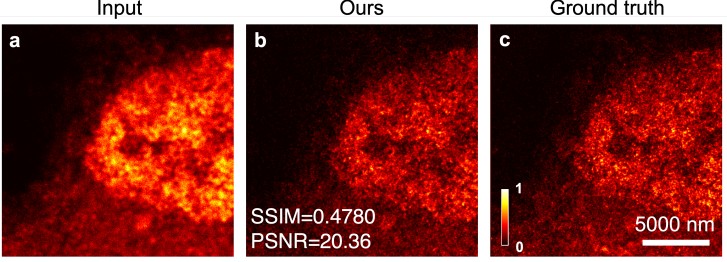

Figure 4: **Microscopy image super-resolution of our method on HeLa cells.** (a) LR confocal image, (b) SR output image and (c) HR STED image of the same FOV.

As illustrated in Fig. 3(a-d), our method effectively reconstructs super-resolved nanobeads beyond the optical diffraction limit and outperforms CMSR in terms of SSIM and PSNR. Moreover, as shown in Fig. 3(e-l), our method reconstructs the nanobeads with smaller diameters, better contrast, and more accurate intensity values, matching well with the ground truth STED images. As another demonstration, we apply our method to fluorescence imaging of HeLa cells and present Fig. 4, Appendix Fig. 10, 11 to further support its success in microscopy image restoration.

### 4.3.2 EVALUATION ON STANDARD IMAGE DATASETS

To further verify the generalization and scalability of our MSCGM, we conducted more experiments on three different standard natural image datasets.

**Natural Image Super-resolution:** We train the three methods on the DIV2K training dataset and then test them on the DIV2K validation set, Set5, and Set14. Table 1 quantifies the super-resolution performance in terms of PSNR and SSIM on the three test sets. The reported MSCGM scores indicate better restoration quality over competitive methods. Table 1 also presents the sampling time of each method under the same resource cost (GPU memory). Our method (MSCGM) achieves superior image generation results and exhibits a 10x improvement in average single-image generation time compared to IR-SDE, an 8x improvement compared to ReFusion, and a 16x improvement compared to BBDM. Additional visualization results of the three methods on the DIV2K validation set are displayed in Fig. 5a, showing better reconstruction fidelity and perceptual quality.

Moreover, we implement fast sampling with fewer sampling steps on the same super-resolution tasks, from 4 to 1000 steps, and depict the results in Fig. 5b. Additional comparative samples, showcasing

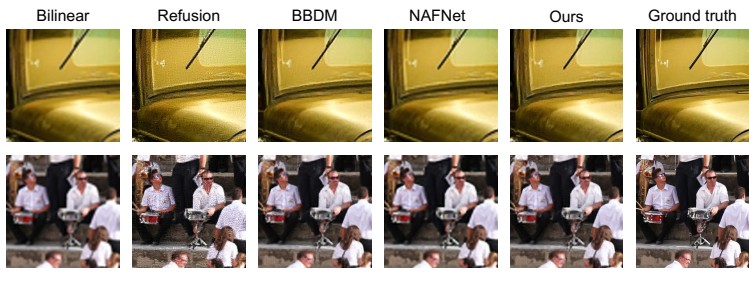

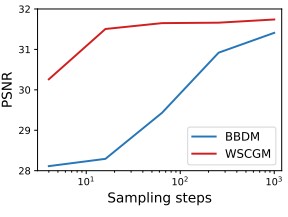

(a) **Comparison of our method and some baselines on** $4\times$ **natural image super-resolution on DIV2K validation set.** Sample steps are set as 1000 for all methods. Center crops of $256 \times 256$ are shown.

(b) **Fast sampling comparison between our method (MSCGM) and BBDM on** $4\times$ **super-resolution.** Use center crops of $256 \times 256$.

Figure 5: Visualization and fast sampling results on natural image super-resolution

the performance of our model against BBDM at fewer sampling steps, are provided in Appendix Figs. 12 and 13.

**Natural Image Shadow Removal:** For image shadow removal task, we train our paradigm on the ISTD training dataset Wang et al. (2018a) of 1330 image triplets (shadow image, mask and clean image) and evaluate it on the ISTD test set of 540 triplets. Appendix Table 4 summarizes the performance of our method against competitive methods, including DC-ShadowNet Jin et al. (2023), ST-CGAN Wang et al. (2018b), DSC Hu et al. (2019), DHAN Cun et al. (2020), BMNet Zhu et al. (2022), ShadowFormer Guo et al. (2023) and BBDM Li et al. (2023). Appendix Fig. 9 further show the visualization results of our method and competitive methods. In summary, our method effectively restores high-quality images from shadowed and low-light images while minimizing the artifacts and inconsistency in the output images.

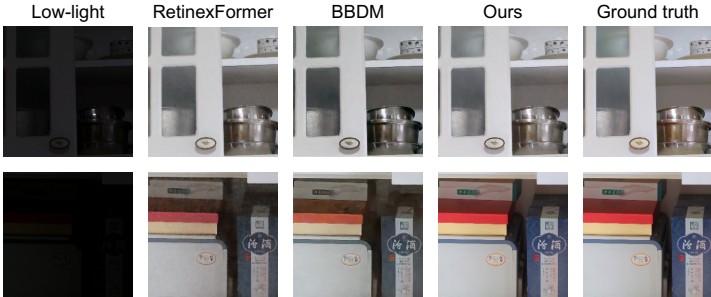

Figure 6: **Image enhancement results on LOL dataset.** Center-cropped $256 \times 256$ patches shown.

**Low-light Natural Image Enhancement:** In this task, we train and evaluate our paradigm on LOL dataset Wei et al. (2018), which contains 485 training and 15 testing image pairs. Figure 6 qualitatively showcases the results of our method in comparison with RetinexFormer Cai et al. (2023) and BBDM. Our method shows better image enhancement performance, and the restored images have more natural colors and less noise details. This effectiveness of our method on image enhancement is further confirmed by the quantitative evaluation results against RetinexFormer, HWMNet Fan et al. (2022) and BBDM, as shown in Appendix Table 3.

## 5 CONCLUSION

To address the limitations of exsiting diffusion models in conditional image restoration, we demonstrate a novel generative model for image restoration based on Brownian bridge process and multiscale wavelet transform. By factorizing the image restoration process in the multi-scale wavelet domains, we utilize Brownian bridge diffusion process and generative adversarial networks to recover different wavelet subbands according to their distribution properties, consequently accelerate the sampling speed significantly and achieve high sample quality and diversity competitive to diffusion model baselines. Future implementation could integrate standard acceleration techniques for diffusion models.

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

## A  CODE AND DATA AVAILABILITY

The codes of our reported method is available at `https://anonymous.4open.science/r/MSCGM-E114`. The DIV2K dataset Agustsson & Timofte (2017) is obtained from `https://data.vision.ee.ethz.ch/cvl/DIV2K/`, and the ISTD dataset Wang et al. (2018a) is obtained from `https://github.com/DeepInsight-PCALab/ST-CGAN`. The microscopy image datasets (nanobeads and HeLa cells) are requested from Wang et al. Wang et al. (2019) and partial demo images are uploaded to `https://anonymous.4open.science/r/MSCGM-E114`.

## B  SCORE REGULARITY FOR DISCRETIZATION

**Theorem 5.** *Suppose the Gaussian distribution $p = N(0, \Sigma)$ and distribution $\tilde{p}_0$ from time reversed SDE, the Kullback-Leibler divergence between $p$ and $p_{\tilde{0}}$ relates to the covariance matrix $\Sigma$ as:*
$KL(p \parallel \tilde{p}_0) \le \Psi_T + \Psi_{\Delta t} + \Psi_{T, \Delta t}$, *with:*

$$\Psi_T = f\left(e^{-4T} \left|\text{Tr}\left((\Sigma - \text{Id})\Sigma\right)\right|\right), \tag{20}$$

$$\Psi_{\Delta t} = f\left(\Delta t \left|\text{Tr}\left(\Sigma^{-1} - \Sigma(\Sigma - \text{Id})^{-1}\log(\Sigma)/2 + (\text{Id} - \Sigma^{-1})/3\right)\right|\right), \tag{21}$$

$$\Psi_{T, \Delta t} = o(\Delta t + e^{-4T}), \qquad \Delta t \to 0, T \to +\infty \tag{22}$$

*where $f(t) = t - \log(1 + t)$ and $d$ is the dimension of $\Sigma$, $\text{Tr}(\Sigma) = d$.*

**Proposition 1.** *For any $\epsilon > 0$, there exists $T, \Delta t \ge 0$ such that:*

$$(1/d)(\Psi_T + \Psi_{\Delta t}) \le \epsilon, \tag{23}$$

$$T/\Delta t \le C\epsilon^{-2}\kappa^3, \tag{24}$$

*where $C \ge 0$ is a universal constant, and $\kappa$ is the condition number of $\Sigma$.*

Guth et al. (2022) provides the proof outline for Theorem 5, based on the following Theorem 6,

**Theorem 6.** *Let $N \in \mathbb{N}$, $\Delta t > 0$, and $T = N\Delta t$. Then, we have that $\bar{x}_t^N \sim N(\hat{\mu}_N, \Sigma^{\widehat{N}})$ with*

$$\Sigma^{\widehat{N}} = \Sigma + \exp(-4T)\Sigma^{\widehat{T}} + \Delta t \Psi^{\widehat{T}} + (\Delta t)^2 R^{\widehat{T}, \Delta t}, \tag{25}$$

$$\hat{\mu}_N = \mu + \exp(-2T)\hat{\mu}_T + \Delta t e^{\widehat{T}} + \frac{(\Delta t)^2}{2} r^{T, \Delta t}, \tag{26}$$

*where $\Sigma^{\widehat{T}}, \Psi^{\widehat{T}}, R^{T, \Delta t} \in \mathbb{R}^{d \times d}$, $\hat{\mu}_T, e^{\widehat{T}}, r^{T, \Delta t} \in \mathbb{R}^d$, and $\|R^{T, \Delta t}\| + \|r^{T, \Delta t}\| \le R$, not dependent on $T \ge 0$ and $\Delta t > 0$. We have that*

$$\Sigma^{\widehat{T}} = -(\Sigma - \text{Id})(\Sigma\Sigma^{-1})^2, \tag{27}$$

$$\Psi^{\widehat{T}} = \text{Id} - \frac{1}{2}\Sigma^2(\Sigma - \text{Id})^{-1}\log(\Sigma) + \exp(-2T)\Psi^{\widetilde{T}}. \tag{28}$$

*In addition, we have*

$$\hat{\mu}_T = -\Sigma^{-1}T\Sigma\mu, \tag{29}$$

$$e^{\widehat{T}} = \left\{-2\Sigma^{-1} - \frac{1}{4}\Sigma(\Sigma - \text{Id})^{-1}\log(\Sigma)\right\}\mu + \exp(-2T)\widetilde{\mu}_T, \tag{30}$$

*with $\Psi^{\widetilde{T}}, \widetilde{\mu}_T$ bounded and not dependent on $T$.*

**Theorem 7.** *Suppose that $\nabla \log p_t(x)$ is $\varphi^2$ in both $t$ and $x$ such that:*

$$\sup_{x, t}\left\|\nabla^2 \log p_t(x)\right\| \le K, \qquad \|\partial_t \nabla \log p_t(x)\| \le Me^{-\alpha t}\|x\| \tag{31}$$

*for some $K$, $M$, $\alpha > 0$. Then, $\|p - \tilde{p}_0\|_{TV} \le \Psi_T + \Psi_{\Delta t} + \Psi_{T, \Delta t}$, where:*

$$\Psi_T = \sqrt{2}e^{-T} \text{KL}\left(p \parallel N(0, \text{Id})\right)^{1/2} \tag{32}$$

$$\Psi_{\Delta t} = 6\sqrt{\Delta t}\left[1 + \mathbb{E}_p\left(\|x\|^4\right)^{1/4}\right]\left[1 + K + M\left(1 + 1/2\alpha\right)^{1/2}\right] \tag{33}$$

$$\Psi_{T, \Delta t} = o\left(\sqrt{\Delta t} + e^{-T}\right) \qquad \Delta t \to 0, T \to +\infty \tag{34}$$

Theorem 7 generalizes Theorem 5 to non-Gaussian processes. Please refer to Guth et al. (2022) for the complete proof.

## C CHARACTERISTICS OF HIGH AND LOW FREQUENCY COEFFICIENTS IN THE WAVELET DOMAIN

### C.1 GAUSSIAN TENDENCY OF LOW-FREQUENCY COEFFICIENTS IN HIGHER SCALES

In an image, pixel intensities are represented as random variables, with adjacent pixels exhibiting correlation due to their spatial proximity. This correlation often follows a power-law decay:

$$C(d) = \frac{1}{(1 + \alpha d)^\beta},\tag{35}$$

where $C(d)$ is the correlation between pixels separated by distance $d$, and $\alpha$ and $\beta$ characterize the rate of decay.

The wavelet transform (i.e., Haar wavelet transform), particularly its down-sampling step, increases the effective distance $d$ among pixels, thereby reducing their original spatial correlation. This reduction is crucial for applying the generalized Central Limit Theorem Rosenblatt (1956); Withers (1981); Ekström (2014); Ash & Doléans-Dade (2000), which requires that the individual variables (pixels, in this case) are not strongly correlated.

At scale $k$ in the wavelet decomposition, the low-frequency coefficients, $\bar{X}_k$, representing the average intensity over $n_k$ pixels, are calculated as:

$$\bar{X}_k = \frac{1}{n_k}(X_1 + X_2 + \ldots + X_{n_k}),\tag{36}$$

where $n_k$ is the number of pixels in each group at scale $k$.

As the scale increases, the effect of averaging over larger groups of pixels, combined with the reduced correlation due to down-sampling, leads to a scenario where the generalized Central Limit Theorem can be applied. Consequently, the distribution of $\bar{X}_k$ tends towards a Gaussian distribution:

$$\bar{X}_k \xrightarrow{\text{d}} N(\mu_k, \frac{\sigma_k^2}{n_k}),\tag{37}$$

where $\mu_k$ and $\sigma_k^2$ are the mean and variance of the averaged intensities at scale $k$, respectively. This Gaussian tendency becomes more pronounced at higher scales due to the combination of reduced pixel correlation and the averaging process.

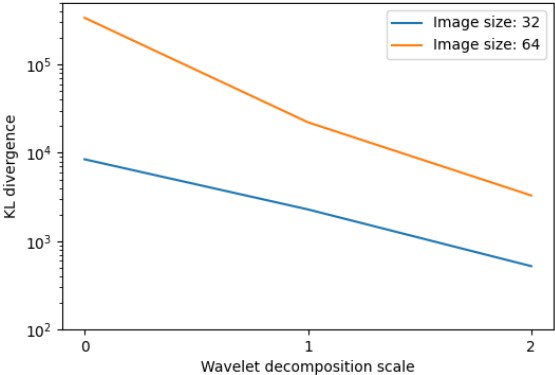

Figure 7: KL divergence between the standard normal distribution and normalized sample distribution with respect to the wavelet scale. Images were sampled from DIV2K dataset by $32 \times 32$ and $64 \times 64$ patches.

In Fig. 7, we quantify the Gaussianity of low-frequency wavelet subbands at different scales using the Kullback-Leibler (KL) divergence, which measures the distance between the standard normal distribution and normalized sample distribution of the low-frequency wavelet coefficients. We

sampled random patches of different resolutions ($32 \times 32$ and $64 \times 64$) from DIV2K dataset to calculate KL divergence. With the increasing of the scale, KL divergence decreases, validating the Gaussian tendency of low-frequency subbands after multi-scale wavelet transforms. Figure 8 further validates this tendency by plotting the kurtosis of sample distribution of microscopy images of nanobeads with respect to the scale.

### C.2 SPARSITY AND NON-GAUSSIANITY OF HIGH-FREQUENCY COEFFICIENTS

High-frequency coefficients, when analyzed through wavelet transform, exhibit a distinct property of sparsity, characterized by a majority of wavelet coefficients being near or at zero, with only a sparse representation of significant non-zero coefficients. This sparsity highlights the efficiency of wavelet transforms in encoding signal details and abrupt changes. Furthermore, these high-frequency components often deviate from Gaussian distributions, tending towards leptokurtic distributions Fraser et al. (2001) with higher peaks and heavier tails. This non-Gaussian nature suggests a concentration of energy in fewer coefficients and is crucial in applications like signal denoising and compression, where recognizing and preserving these vital characteristics is paramount.

In the following proposition, we theoretically show that the conditional distribution of $\boldsymbol{x}_H^k$ on $\boldsymbol{x}_L^k$ exhibits highly non-Gaussian properties and yields sparse samples. For a given image $\boldsymbol{x}$ and threshold $t$, the sparsity of its high-frequency coefficients at $k$-scale is defined as:

$$s(\boldsymbol{x}_H^k) = \frac{\|\mathbf{1}\{\boldsymbol{x}_H^k \leq t\}\|}{L^2}, \ k = 1, 2, \ldots \tag{38}$$

Here $\|\cdot\|$ is the norm counting the number of 1s in the vector. In this way, we could estimate the expected sparsity of the true marginal distribution $p(\boldsymbol{x}_H^k)$. Considering that the LL coefficients with approximate Gaussian distribution given the whitening effect of wavelet decomposition, we have the following proposition.

**Proposition 2.** *For a sufficiently large $k$, if the expected sparsity of $\boldsymbol{x}_H^k$ has a lower bound $\alpha$*

$$\mathbb{E}(s(\boldsymbol{x}_H^k)) \geq \alpha, \tag{39}$$

*where $\alpha \in [0, 1]$. Then the conditional expected sparsity of $\boldsymbol{x}_H^k$ on $\boldsymbol{x}_L^k$ is bounded by*

$$\mathbb{E}(s(\boldsymbol{x}_H^k)|\boldsymbol{x}_L^k) \geq \alpha - \varepsilon, \tag{40}$$

*where $\varepsilon > 0$ is a small positive number determined by $k$.*

*Proof.* According to Eq. 37, for a sufficiently large $k$ we could assume that

$$\int |p(\boldsymbol{x}_L^k) - f_k(\boldsymbol{x}_L^k)|d\boldsymbol{x}_L^k \leq \varepsilon, \tag{41}$$

where $f_k(\boldsymbol{x}_L^k)$ is the PDF of standard Gaussian distribution. Notice that

$$\mathbb{E}(s(\boldsymbol{x}_H^k)) = \iint s(\boldsymbol{x}_H^k)p(s(\boldsymbol{x}_H^k)|\boldsymbol{x}_L^k)p(\boldsymbol{x}_L^k)d\boldsymbol{x}_L^k ds \tag{42}$$

$$= \int \mathbb{E}(s(\boldsymbol{x}_H^k)|\boldsymbol{x}_L^k)p(\boldsymbol{x}_L^k)d\boldsymbol{x}_L^k \geq \alpha \tag{43}$$

Since $s$ is a bounded function in $[0, 1]$, $\mathbb{E}(s(\boldsymbol{x}_H^k)|\boldsymbol{x}_L^k)$ has an uniform lower bound with respect to all $\boldsymbol{x}_L^k$, denoted as $\alpha'$. In other words, there exists $\alpha' \in [0, 1]$ such that

$$\mathbb{E}(s(\boldsymbol{x}_H^k)|\boldsymbol{x}_L^k) \geq \alpha', \ \forall \boldsymbol{x}_L^k \tag{44}$$

We can get

$$\int \mathbb{E}(s(\boldsymbol{x}_H^k)|\boldsymbol{x}_L^k)p(\boldsymbol{x}_L^k)d\boldsymbol{x}_L^k$$

$$= \int \mathbb{E}(s(\boldsymbol{x}_H^k)|\boldsymbol{x}_L^k)f_k(\boldsymbol{x}_L^k)d\boldsymbol{x}_L^k \tag{45}$$

$$+ \int \mathbb{E}(s(\boldsymbol{x}_H^k)|\boldsymbol{x}_L^k)(p(\boldsymbol{x}_L^k) - f_k(\boldsymbol{x}_L^k))d\boldsymbol{x}_L^k \geq \alpha'$$

Similarly, it is easy to see that $1$ is a trivial uniform upper bound for $\mathbb{E}(s(\boldsymbol{x}_H^k)|\boldsymbol{x}_L^k)$. Thus,

$$\mathbb{E}(s(\boldsymbol{x}_H^k)|\boldsymbol{x}_L^k) \geq \alpha' = \int \mathbb{E}(s(\boldsymbol{x}_H^k)|\boldsymbol{x}_L^k)p(\boldsymbol{x}_L^k)d\boldsymbol{x}_L^k - \varepsilon \geq \alpha - \varepsilon. \tag{46}$$

$\square$

### C.3 QUANTIFYING NON-GAUSSIANITY OF DATASETS

Third- and fourth-order sample cumulants, i.e., skewness and kurtosis, to quantify the non-Gaussianity of certain sample distributions Groeneveld & Meeden (1984). The non-Gaussianity of high-frequency subbands can be evidenced by the kurtosis plot with respect to wavelet scales in Fig. 8. The kurtosis of high-frequency subbands of microscopy images increases with the wavelet scales, showing the high non-Gaussianity of the distribution of high-frequency coefficients.

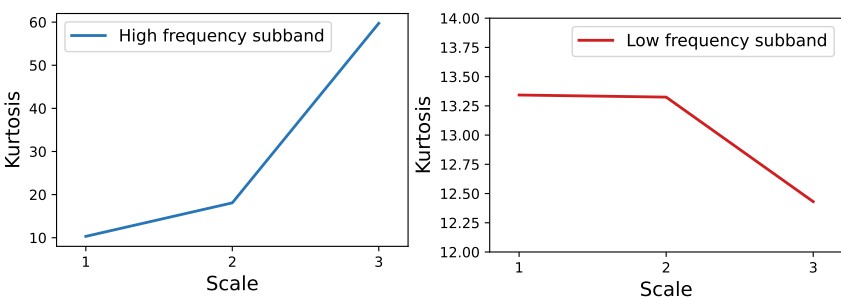

Figure 8: Kurtosis of (left) low-frequency and (right) high-frequency subband coefficients with respect to the wavelet scale. Metrics were calculated on microscopy images of nanobeads.

In the following subsections, we first introduce the definitions of the two metrics and then evaluate and compare the non-Gaussianity of DIV2K and microscopy nanobead datasets.

#### C.3.1 SKEWNESS ($\gamma_1$)

Skewness is a measure of the asymmetry of the probability distribution of a real-valued random variable. It quantifies how much the distribution deviates from a normal distribution in terms of asymmetry. The skewness value can be positive, zero, negative, or undefined. In a perfectly symmetrical distribution, skewness is zero. Positive skewness indicates a distribution with an extended tail on the right side, while negative skewness shows an extended tail on the left side.

The mathematical formula for skewness is given by:

$$\gamma_1 = E\left[\left(\frac{X - \mu}{\sigma}\right)^3\right] \tag{47}$$

where $X$ is the random variable, $\mu$ is the mean of $X$, $\sigma$ is the standard deviation of $X$, and $E$ denotes the expected value.

The greater the absolute value of the skewness, the higher the degree of non-Gaussianity in the distribution.

#### C.3.2 KURTOSIS ($\beta_2$)

Kurtosis is a measure of the "tailedness" of the probability distribution of a real-valued random variable. It provides insights into the shape of the distribution's tails and peak. High kurtosis in a data set suggests a distribution with heavy tails and a sharper peak (leptokurtic), while low kurtosis indicates a distribution with lighter tails and a more flattened peak (platykurtic). Kurtosis is often compared to the normal distribution, which has a kurtosis of 3 (excess kurtosis of 0).

The formula for kurtosis is:

$$\beta_2 = E\left[\left(\frac{X - \mu}{\sigma}\right)^4\right] - 3 \tag{48}$$

where the variables represent the same as in the skewness formula.

These statistical measures, skewness ($\gamma_1$) and kurtosis ($\beta_2$), are crucial for quantifying and analyzing the non-Gaussianity in image data. They provide valuable insights into the distribution characteristics of image pixel intensities, particularly in highlighting deviations from the normal distribution.

The higher the kurtosis, the greater the degree of non-Gaussianity in the distribution, indicating a distribution with heavier tails than a normal distribution.

### C.3.3 NON-GAUSSIANITY OF DATASETS

Here we examine the non-Gaussianity of the distribution of DIV2K training dataset and the microscopy nanobead dataset used in this work. Table 2 summarizes their skewnesses and kurtoses. We observe that microscopy images tend to have larger absolute values of skewness and kurtosis, confirming their highly non-Gaussian distribution and the high condition number for microscopy image restoration problem. As a result, standard DMs with the assumption that the distribution of target images is close to normal does not hold for microscopy images and may lead to performance degradation and excessive sampling time.

| Dataset | Skewness | Kurtosis |
|---|---|---|
| DIV2K | 0.2193 | -1.0373 |
| Microscopy nanobeads | 2.5852 | 13.3429 |

Table 2: Quantitative measure of non-Gaussianity of DIV2K and microscopy image datasets.

## D  WAVELET TRANSFORM

Wavelet transforms are derived from a single prototype function known as the 'mother wavelet'. This function undergoes various scaling and translation processes to generate a family of wavelets. The general form of a wavelet function $\psi(x)$, derived from the mother wavelet, is expressed as:

$$\psi_{a,b}(x) = \frac{1}{\sqrt{|a|}}\psi\left(\frac{x-b}{a}\right),\tag{49}$$

where $a, b \in \mathbb{R}$, $a \neq 0$, and $x \in D$, with $a$ and $b$ representing scaling and translation parameters, respectively, and $D$ being the domain of the wavelets.

### D.1  MULTIRESOLUTION ANALYSIS (MRA)

MRA is crucial in the Wavelet Transform Jawerth & Sweldens (1994); Ranta (2010). It involves constructing a basic functional basis in a subspace $V_0$ within $L^2(\mathbb{R})$ and then expanding this basis through scaling and translation to cover the entire space $L^2(\mathbb{R})$ for multiscale analysis.

For each $k \in \mathbb{Z}$ and $k \in \mathbb{N}$, we define the scale function space as $\mathbf{V}_k = \{f | f$ is restricted to $(2^{-k}l, 2^{-k}(l+1))$ for all $l = 0, 1, \ldots, 2^k - 1$ and vanishes elsewhere$\}$. Each space $\mathbf{V}_k$ encompasses a dimension of $2^k$, with each subspace $\mathbf{V}_i$ nested within $\mathbf{V}_{i+1}$:

$$\mathbf{V}_0 \subseteq \mathbf{V}_1 \subseteq \mathbf{V}_2 \subseteq \ldots \subseteq \mathbf{V}_n \subseteq \ldots.\tag{50}$$

Using a base function $\varphi(x)$ in $\mathbf{V}_0$, we can span $\mathbf{V}_k$ with $2^n$ functions derived from $\varphi(x)$ through scaling and translation:

$$\varphi_l^k(x) = 2^{k/2}\varphi(2^k x - l), \quad l = 0, 1, \ldots, 2^k - 1.\tag{51}$$

These functions, $\varphi_l(x)$, are known as scaling functions and they project any function into the approximation space $\mathbf{V}_0$. The orthogonal complement of $\mathbf{V}_k$ in $\mathbf{V}_{k+1}$ is the wavelet subspace $\mathbf{W}_k$, satisfying:

$$\mathbf{V}_k \oplus \mathbf{W}_k = \mathbf{V}_{k+1}, \quad \mathbf{V}_k \perp \mathbf{W}_k.\tag{52}$$

Thus, we can decompose $\mathbf{V}_k$ as:

$$\mathbf{V}_k = \mathbf{V}_0 \oplus \mathbf{W}_0 \oplus \mathbf{W}_1 \oplus \ldots \oplus \mathbf{W}_{k-1}.\tag{53}$$

To form the orthogonal basis for $\mathbf{W}_k$, we construct it within $L^2(\mathbb{R})$. This basis is derived from the wavelet function $\psi(x)$, orthogonal to the scaling function $\varphi(x)$. The wavelet function is defined as:

$$\psi_l^k(x) = 2^{k/2}\psi(2^k x - l), \quad l = 0, 1, \ldots, 2^k - 1. \tag{54}$$

A key property of these wavelet functions is their orthogonality to polynomials of lower order, exemplified by the vanishing moments criterion for first-order polynomials:

$$\int_{-\infty}^{\infty} x\psi_j(x)dx = 0. \tag{55}$$

Orthogonality and vanishing moments are central to wavelets, enabling efficient data representation and feature extraction by capturing unique data characteristics without redundancy. This efficiency is particularly useful in areas like signal processing and image compression.

## D.2 WAVELET DECOMPOSITION AND RECONSTRUCTION

The Discrete Wavelet Transform (DWT) provides a multi-resolution analysis of signals, useful in various applications Gupta et al. (2021; 2022); Xiao et al. (2023). For a discrete signal $f[n]$, DWT decomposes it into approximation coefficients $cA$ and detail coefficients $cD$ using the scaling function $\varphi(x)$ and the wavelet function $\psi(x)$, respectively:

$$cA[k] = \sum_n h[n - 2k]f[n], \tag{56}$$

$$cD[k] = \sum_n g[n - 2k]f[n], \tag{57}$$

where $h[n]$ and $g[n]$ are the low-pass and high-pass filters, respectively. This decomposition process can be recursively applied for deeper multi-level analysis.

Reconstruction of the signal $f'[n]$ from its wavelet coefficients uses inverse transformations, employing synthesis filters $h_0[n]$ and $g_0[n]$:

$$f'[n] = \sum_k cA[k] \cdot h_0[n - 2k] + cD[k] \cdot g_0[n - 2k], \tag{58}$$

where the synthesis filters are typically the time-reversed counterparts of the decomposition filters. In multi-level decompositions, reconstruction is a stepwise process, beginning with the coarsest approximation and progressively incorporating higher-level details until the original signal is reconstructed.

## D.3 HAAR WAVELET TRANSFORM

The Haar wavelet Stanković & Falkowski (2003), known for its simplicity and orthogonality, is a fundamental tool in digital signal processing. Its straightforward nature makes it an ideal choice for a variety of applications, which is why we have incorporated it into our project. The discrete wavelet transform (DWT) using Haar wavelets allows for the efficient decomposition of an image into a coarse approximation of its main features and detailed components representing high-frequency aspects. This process, enhanced by multi-resolution analysis (MRA), facilitates the examination of the image at various scales, thereby uncovering more intricate details.

The mathematical representation of the Haar wavelet and its scaling function is as follows:

$$\psi(t) = \begin{cases} 1 & \text{if } 0 \le t < 0.5, \\ -1 & \text{if } 0.5 \le t < 1, \\ 0 & \text{otherwise}, \end{cases} \tag{59}$$

$$\phi(t) = \begin{cases} 1 & \text{if } 0 \le t < 1, \\ 0 & \text{otherwise}, \end{cases} \tag{60}$$

where $\psi(t)$ denotes the Haar wavelet function and $\phi(t)$ is the corresponding scaling function.

The application of Haar wavelet transform extends to two-dimensional spaces, particularly in image processing. This extension utilizes the same decomposition approach as for one-dimensional signals but applies it to both rows and columns of the image. The filter coefficients for Haar wavelets are calculated through these inner product evaluations:

- Low-pass filter coefficients (h):

$$h_0 = \int_0^1 \phi(t) \cdot \phi(2t)\, dt,$$
$$h_1 = \int_0^1 \phi(t) \cdot \phi(2t - 1)\, dt. \tag{61}$$

- High-pass filter coefficients (g):

$$g_0 = \int_0^1 \psi(t) \cdot \psi(2t)\, dt,$$
$$g_1 = -\int_0^1 \psi(t) \cdot \psi(2t - 1)\, dt. \tag{62}$$

To normalize these coefficients (ensuring their L2 norm is 1), we find $h_0 = \frac{1}{\sqrt{2}}$ and similar values for the other coefficients. Thus, the Haar filter coefficients are:

$$h = \left[\frac{1}{\sqrt{2}}, \frac{1}{\sqrt{2}}\right], \qquad g = \left[\frac{1}{\sqrt{2}}, -\frac{1}{\sqrt{2}}\right]. \tag{63}$$

For two-dimensional DWT in image processing, these filter coefficients are matrix-operated on the image. The horizontal operation uses the outer product of the filter vector with a column vector, and the vertical operation uses the outer product with a row vector. The resulting filter matrices are:

$$H = h^T \otimes h = \begin{bmatrix} \frac{1}{\sqrt{2}} & \frac{1}{\sqrt{2}} \\ \frac{1}{\sqrt{2}} & \frac{1}{\sqrt{2}} \end{bmatrix},$$
$$G = g^T \otimes g = \begin{bmatrix} \frac{1}{\sqrt{2}} & -\frac{1}{\sqrt{2}} \\ -\frac{1}{\sqrt{2}} & \frac{1}{\sqrt{2}} \end{bmatrix}. \tag{64}$$

The matrix $H$ corresponds to low-pass filtering (approximation), while $G$ captures the high-frequency details. In image transformation using Haar wavelets, these matrices help derive various coefficients representing different aspects of the image, such as approximation($LL$), horizontal($LH$), vertical($HL$), and diagonal detail($HH$) components.

# E    DUALITY PROOF

## E.1    GENERATIVE MODELING IN SPATIAL DOMAIN

For the Score-based Generative Model (SGM) Song et al. (2020b); Ho et al. (2020); Song & Ermon (2019), the forward/noising process is mathematically formulated as the Ornstein-Uhlenbeck (OU) process. The general time-rescaled OU process is expressed as:

$$d\boldsymbol{X}_t = -g(t)^2 \boldsymbol{X}_t dt + \sqrt{2}g(t)d\boldsymbol{B}_t. \tag{65}$$

Here, $(\boldsymbol{X}_t)_{t \in [0,T]}$ represents the noising process starting with $\boldsymbol{X}_0$ sampled from the data distribution. $(\boldsymbol{B}_t)_{t \in [0,T]}$ denotes a standard d-dimensional Brownian motion. The reverse process, $\boldsymbol{X}_t^{\leftarrow}$, is defined such that $(\boldsymbol{X}_t^{\leftarrow})_{t \in [0,T]} = (\boldsymbol{X}_{T-t})_{t \in [0,T]}$. Assuming $g(t) = 1$ in standard diffusion models, the reverse process is:

$$d\boldsymbol{X}_t^{\leftarrow} = \left(\boldsymbol{X}_t^{\leftarrow} + 2\nabla \log p_{T-t}(\boldsymbol{X}_t^{\leftarrow})\right) dt + \sqrt{2}d\boldsymbol{B}_t. \tag{66}$$

Here, $p_t$ is the marginal density of $\boldsymbol{X}_t$, and $\nabla \log p_t$ is the score. To revert $\boldsymbol{X}_0$ from $\boldsymbol{X}_T$ via the time-reversed SDE, accurate estimation of the score $\nabla \log p_t$ at each time $t$ is essential, alongside minimal error introduction during SDE discretization.

The reverse process approximation, as specified in Eq. 66, involves time discretization and score approximation $\nabla \log p_t$ by $s_t$, forming a Markov chain approximation of the time-reversed SDE. The chain starts with $\tilde{\mathbf{x}}_T \sim N(0, I_d)$, evolving over uniform time intervals $\Delta t$, from $t_N = T$ to $t_0 = 0$. The discretized process is detailed as:

$$\tilde{\boldsymbol{x}}_{t-1} = \tilde{\boldsymbol{x}}_t + \Delta t \left(\tilde{\boldsymbol{x}}_t + 2s_t(\tilde{\boldsymbol{x}}_t)\right) + \sqrt{2\Delta t}\mathbf{z}_t, \tag{67}$$

where $\mathbf{z}_t$ are instances of Brownian motion $\boldsymbol{B}_t$.

### E.2 GENERATIVE MODELING IN WAVELET DOMAIN

Consider $\mathbf{X}$ as the image vector in the spatial domain. The discrete wavelet transform (DWT) Shensa et al. (1992); Heil & Walnut (1989) of $\mathbf{X}$ can be expressed as:

$$\widehat{\mathbf{X}} = \mathbf{A}\mathbf{X}, \quad \mathbf{X} \in \mathbb{R}^d.$$

Here, $\mathbf{A}$ represents the discrete wavelet transform matrix, which is orthogonal, satisfying $\mathbf{A}\mathbf{A}^\top = \mathbf{I}$. Various forms of $\mathbf{A}$ are utilized in practice, such as Haar wavelets.

In the context of score-based generative modeling, we consider the forward (or noising) process, which can be formulated by the Ornstein–Uhlenbeck (OU) process. This is mathematically described as:

$$d\mathbf{X}_t = -\gamma(t)^2\mathbf{X}_t dt + \sqrt{2}\gamma(t)d\mathbf{B}_t, \tag{68}$$

where $\mathbf{B}_t$ is a standard d-dimensional Brownian motion. Upon applying DWT to $\mathbf{X}_t$, the transformed $\widehat{\mathbf{X}}_t$ also follows the OU process:

$$d\widehat{\mathbf{X}}_t = -\gamma(t)^2\widehat{\mathbf{X}}_t dt + \sqrt{2}\gamma(t)\mathbf{A}d\mathbf{B}_t, \quad \widehat{\mathbf{X}}_0 = \mathbf{A}\mathbf{X}_0. \tag{69}$$

Defining $\widehat{\mathbf{B}}_t = \mathbf{A}\mathbf{B}_t$, which also behaves as a standard Brownian motion. If $\mathbf{X}_0$ is sampled from a distribution $p$, then $\widehat{\mathbf{X}}_0$ originates from the distribution

$$\hat{q} = \mathcal{T}_\mathbf{A}\#p, \tag{70}$$

where $\mathcal{T}_\mathbf{A}$ denotes the linear transformation operation by $\mathbf{A}$ and $\#$ represents the pushforward operation. Consequently, we have

$$\hat{q}(\mathbf{x}) = p(\mathbf{A}^\top\mathbf{x}). \tag{71}$$

Let $p_t$ be the density distribution of $\mathbf{X}_t$ and $\hat{q}_t$ that of $\widehat{\mathbf{X}}_t$. Then,

$$\hat{q}_t = \mathcal{T}_\mathbf{A}\#p_t, \quad \hat{q}_t(\mathbf{x}) = p_t(\mathbf{A}^\top\mathbf{x}). \tag{72}$$

Define the score functions for both processes as:

$$\mathbf{s}_t = \nabla\log p_t, \quad \mathbf{r}_t = \nabla\log\hat{q}_t. \tag{73}$$

These functions are related by:

$$\mathbf{r}_t(\mathbf{x}) = \frac{\nabla\hat{q}_t(\mathbf{x})}{\hat{q}_t(\mathbf{x})} = \frac{\mathbf{A}\nabla p_t(\mathbf{A}^\top\mathbf{x})}{p_t(\mathbf{A}^\top\mathbf{x})} = \mathbf{A}\mathbf{s}_t(\mathbf{A}^\top\mathbf{x}). \tag{74}$$

For the reverse processes denoted as $\mathbf{X}_t^\leftarrow$ and $\widehat{\mathbf{X}}_t^\leftarrow$, assuming $\gamma(t) = 1$, they are given by:

$$\begin{aligned} d\mathbf{X}_t^\leftarrow &= \left(\mathbf{X}_t^\leftarrow + 2\mathbf{s}_{T-t}(\mathbf{X}_t^\leftarrow)\right)dt + \sqrt{2}d\mathbf{B}_t, \\ d\widehat{\mathbf{X}}_t^\leftarrow &= \left(\widehat{\mathbf{X}}_t^\leftarrow + 2\mathbf{r}_{T-t}(\widehat{\mathbf{X}}_t^\leftarrow)\right)dt + \sqrt{2}d\widehat{\mathbf{B}}_t. \end{aligned} \tag{75}$$

Here, $\widehat{\mathbf{B}}_t = \mathbf{A}\mathbf{B}_t$. Exploring the second SDE:

$$\begin{aligned} d\widehat{\mathbf{X}}_t^\leftarrow &= \left(\widehat{\mathbf{X}}_t^\leftarrow + 2\mathbf{r}_{T-t}(\widehat{\mathbf{X}}_t^\leftarrow)\right)dt + \sqrt{2}d\widehat{\mathbf{B}}_t \\ &= \left(\widehat{\mathbf{X}}_t^\leftarrow + 2\mathbf{A}\mathbf{s}_{T-t}(\mathbf{A}^\top\widehat{\mathbf{X}}_t^\leftarrow)\right)dt + \sqrt{2}d\widehat{\mathbf{B}}_t \\ \mathbf{A}^\top d\widehat{\mathbf{X}}_t^\leftarrow &= \left(\mathbf{A}^\top\widehat{\mathbf{X}}_t^\leftarrow + 2\mathbf{s}_{T-t}(\mathbf{A}^\top\widehat{\mathbf{X}}_t^\leftarrow)\right)dt + \sqrt{2}\mathbf{A}^\top d\widehat{\mathbf{B}}_t. \end{aligned} \tag{76}$$

Substituting $\mathbf{A}^\top\widehat{\mathbf{X}}_t^\leftarrow$ with $\mathbf{X}_t^\leftarrow$ brings us back to the first equation. The training processes for $\mathbf{s}\theta, \mathbf{r}\hat{\theta}$ with $\mathbf{X}_t^{(i)}, \widehat{\mathbf{X}}_t^{(i)}$ also follow the standard denoising score matching loss function:

$$\mathbb{E}_t\left\{\lambda(t)\mathbb{E}_{\mathbf{X}_0}\mathbb{E}_{\mathbf{X}_t|\mathbf{X}_0}\left[\|\mathbf{s}_\theta(\mathbf{X}_t, t) - \nabla_{\mathbf{X}_t}\log p_{0t}(\mathbf{X}_t|\mathbf{X}_0)\|^2\right]\right\}$$

$$\mathbb{E}_t\left\{\hat{\lambda}(t)\mathbb{E}_{\widehat{\mathbf{X}}_0}\mathbb{E}_{\widehat{\mathbf{X}}_t|\widehat{\mathbf{x}}_0}\left[\left\|\mathbf{r}_{\hat{\theta}}(\widehat{\mathbf{X}}_t, t) - \nabla_{\widehat{\mathbf{X}}_t}\log\hat{q}_{0t}(\widehat{\mathbf{X}}_t|\widehat{\mathbf{X}}_0)\right\|^2\right]\right\}. \tag{77}$$

The forward and reverse probability distribution functions $p_{0t}$ and $\hat{q}_{0t}$ are defined as per the standard SGM model:

$$p_{0t}(\mathbf{X}_t|\mathbf{X}_0) = N(\mathbf{X}_t; \sqrt{\overline{\alpha}_t}\mathbf{X}_0, (1-\overline{\alpha}_t)\mathbf{I}),$$
$$\hat{q}_{0t}(\widehat{\mathbf{X}}_t|\widehat{\mathbf{X}}_0) = N(\widehat{\mathbf{X}}_t; \sqrt{\overline{\alpha}_t}\widehat{\mathbf{X}}_0, (1-\overline{\alpha}_t)\mathbf{I}). \tag{78}$$

## F  DATASETS AND IMPLEMENTATION DETAILS

For super-resolution experiments on DIV2K, the LR images were bicubically downsampled to $64 \times 64$ center-cropped patches and bilinearly upsampled to $256 \times 256$ as the input images, and the HR images were center-cropped as $256 \times 256$. For shadow removal on the ISTD dataset, we used a similar method, pairing original shadow images with their shadow-free counterparts, both consistently sized to maintain uniformity in processing.

For super-resolution experiments involving HeLa cell nuclei and nano-beads, we utilized a Leica TCS SP8 stimulated emission depletion (STED) confocal microscope, equipped with a Leica HC PL APO 100×/1.40-NA Oil STED White objective. The samples were prepared on high-performance coverslips (170μm ± 10μm, Carl Zeiss Microscopy), facilitating precise imaging. In the staining process, HeLa cell nuclei were treated with Rabbit anti-Histone H3 and Atto-647N Goat anti-rabbit IgG antibodies. The nano-beads, used for the STED experiments, were processed with methanol and ProLong Diamond antifade reagents. Both samples were excited with a 633-nm wavelength laser. Emission detection was carried out using a HyD SMD photodetector (Leica Microsystems) through a 645–752-nm bandpass filter.

For super-resolution experiments on microscopy images of nanobeads, the low-resolution (LR) images were acquired with specific settings (16 times line average and 30 times frame average for nano-beads; 8 times line average and 6 times frame average for cell nuclei), ensuring the acquisition of high-quality images necessary for our computer vision analysis. The scanning step size, i.e., the effective pixel size is 30 nm. Here, all LR and HR images were center cropped as $256 \times 256$ patches to match the setting in other experiments.

AdamW Loshchilov & Hutter (2017) optimizers were used in all experiments with an initial learning rate of 1e-4. Models with exponential moving averaged parameters with a rate of 0.999 was saved and evaluated. The BBDM models at the coarsest wavelet scale were trained for 100000 steps with a batch size of 6.

In our GAN training, we used a batch size of 3, with images cropped at 256x256 resolution. We employed AdamW optimizers, setting the learning rate at 1e-4 for the generator and 1e-5 for the discriminator. The training loss function is a weighted sum of diverse loss terms: L1 loss with a weight of 20.0, adversarial loss at 0.1, and structural similarity index measure (SSIM) Wang et al. (2004) loss weighted at 0.5. The training was conducted for 200 epochs.

All models were implemented on NVIDIA V100 graphic cards using PyTorch. For speed test, a batch size of 16 was used for BBDM, IR-SDE and Refusion models, and a batch size of 64 was used for MSCGM so that they consume equivalent computational resources.

The complete training process of WSCGM is described below:

---
**Algorithm 3** WSCGM Training

---
1: Sample $(\boldsymbol{x}_0, \boldsymbol{y}) \sim q(\boldsymbol{x}_0, \boldsymbol{y})$
2: Wavelet transform $\boldsymbol{x}_0, \boldsymbol{y}$ by $S$ times to get $\{\boldsymbol{x}_L^S, \boldsymbol{x}_H^S, \cdots, \boldsymbol{x}_H^1, \boldsymbol{y}_L^S, \boldsymbol{y}_H^S, \cdots, \boldsymbol{y}_H^1\}$
3: Sample $t \sim \text{Uniform}([1, \cdots, T]), \boldsymbol{\epsilon}_t \sim N(\boldsymbol{0}, \boldsymbol{I})$
4: Take gradient step on $\nabla_\theta \| m_t(\boldsymbol{y}_L^S - \boldsymbol{x}_L^S) + \sqrt{\delta}\boldsymbol{\epsilon}_t - \boldsymbol{\epsilon}_\theta(\boldsymbol{x}_{t,L}^S, \boldsymbol{y}, t) \|^2$
5: Take gradient step on $L_G$ and $L_D$

---

# G ADDITIONAL RESULTS

## G.1 RESULTS ON IMAGE ENHANCEMENT

Here we provide the metrics comparison results of image enhancement.

| Methods | PSNR (dB)↑ | SSIM↑ | FID↓ |
|---|---|---|---|
| BBDM | 28.66 | 0.860 | 98.49 |
| HWMNet | 29.40 | 0.902 | **82.30** |
| RetinexFormer | **29.55** | **0.897** | 109.75 |
| MSCGM | 29.27 | 0.863 | 115.96 |

Table 3: **Comparison of HWMNetFan et al. (2022), RetinexFormerCai et al. (2023) and our method (MSCGM) on image enhancement experiment.** Metrics calculated on $256 \times 256$ center-cropped images of LOL dataset. Entries in bold indicate the best performance achieved among the compared method.

## G.2 RESULTS ON SHADOW REMOVAL TASK

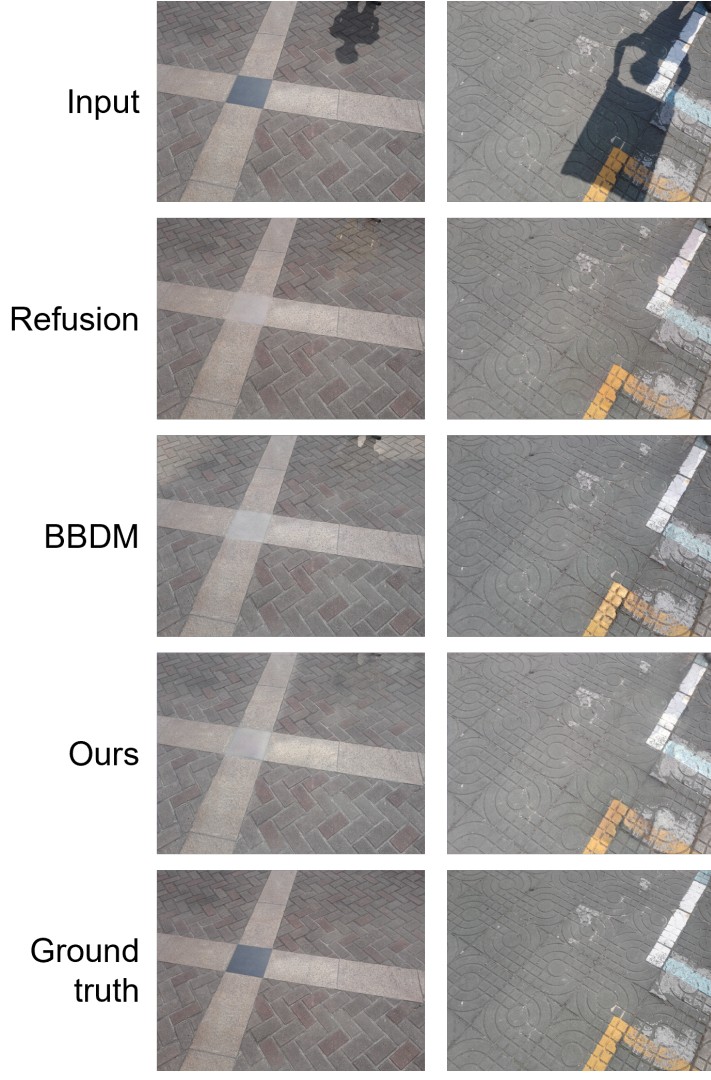

Figure 9: Comparison of our method and baselines on the natural image shadow removal task on ISTD test set. Sample steps were set as 1000 for all methods.

Figure 9 visualizes the shadow removal performance of our approach and competitive methods. Overall, our method generates output images with better consistency and milder artifacts. Refer to Table 4 for quantitative assessments on these results.

| Methods | PSNR (dB)↑ | SSIM↑ |
|---|---|---|
| DC-ShadowNet Jin et al. (2023) | 26.38 | 0.922 |
| ST-CGAN Wang et al. (2018b) | 27.44 | 0.929 |
| DSC Hu et al. (2019) | 30.64 | 0.843 |
| DHANCun et al. (2020) | 27.88 | 0.921 |
| BMNet Zhu et al. (2022) | 30.28 | 0.927 |
| ShadowFormer Guo et al. (2023) | 30.47 | **0.935** |
| BBDM Li et al. (2023) | 30.54 | 0.910 |
| MSCGM(Ours) | **31.08** | 0.915 |

Table 4: **Quantitative evaluation metrics of our method (MSCGM) and competitive methods on ISTD dataset.** Metrics calculated on full-resolution images.

### G.3 ADDITIONAL RESULTS ON MICROSCOPY IMAGE SUPER-RESOLUTION

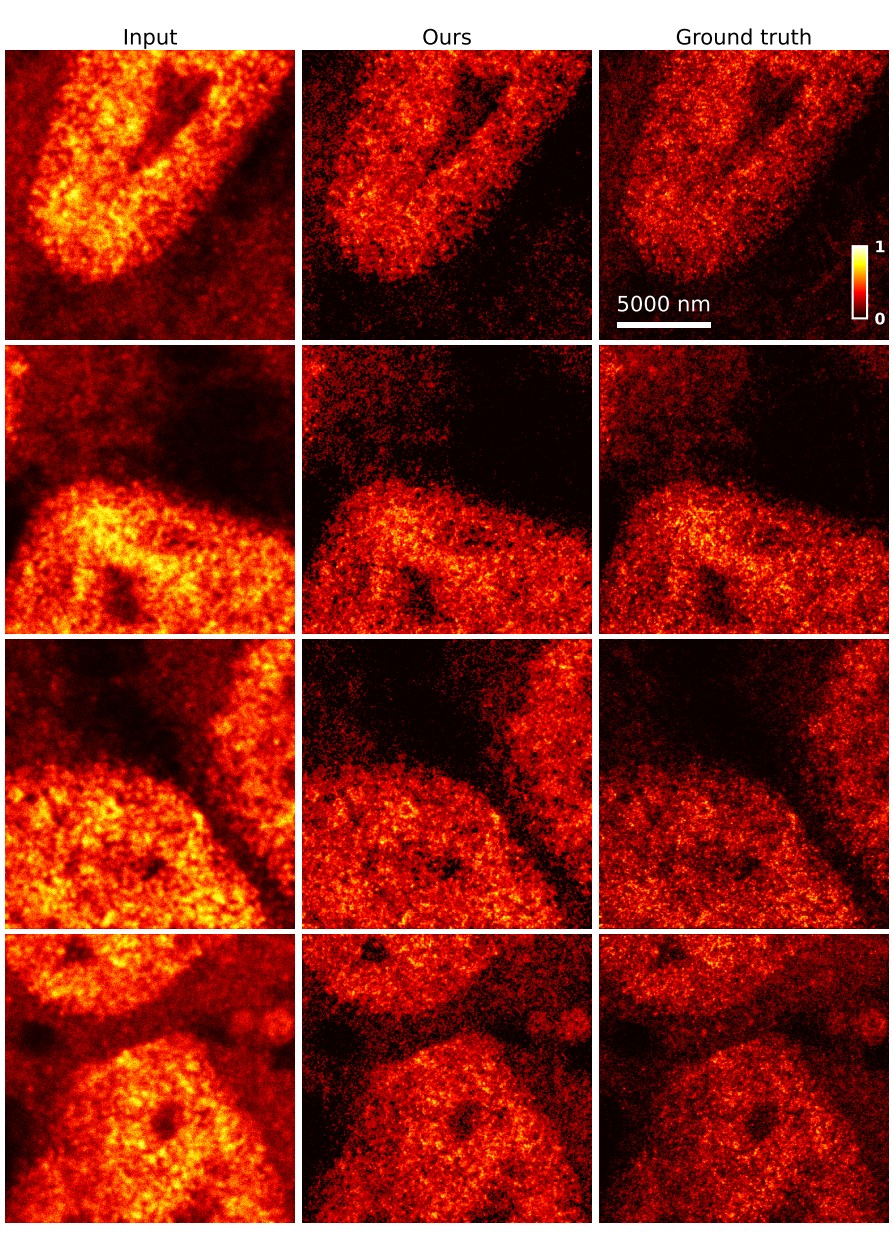

Figure 10: Additional microscopy image super-resolution results of HeLa cells.

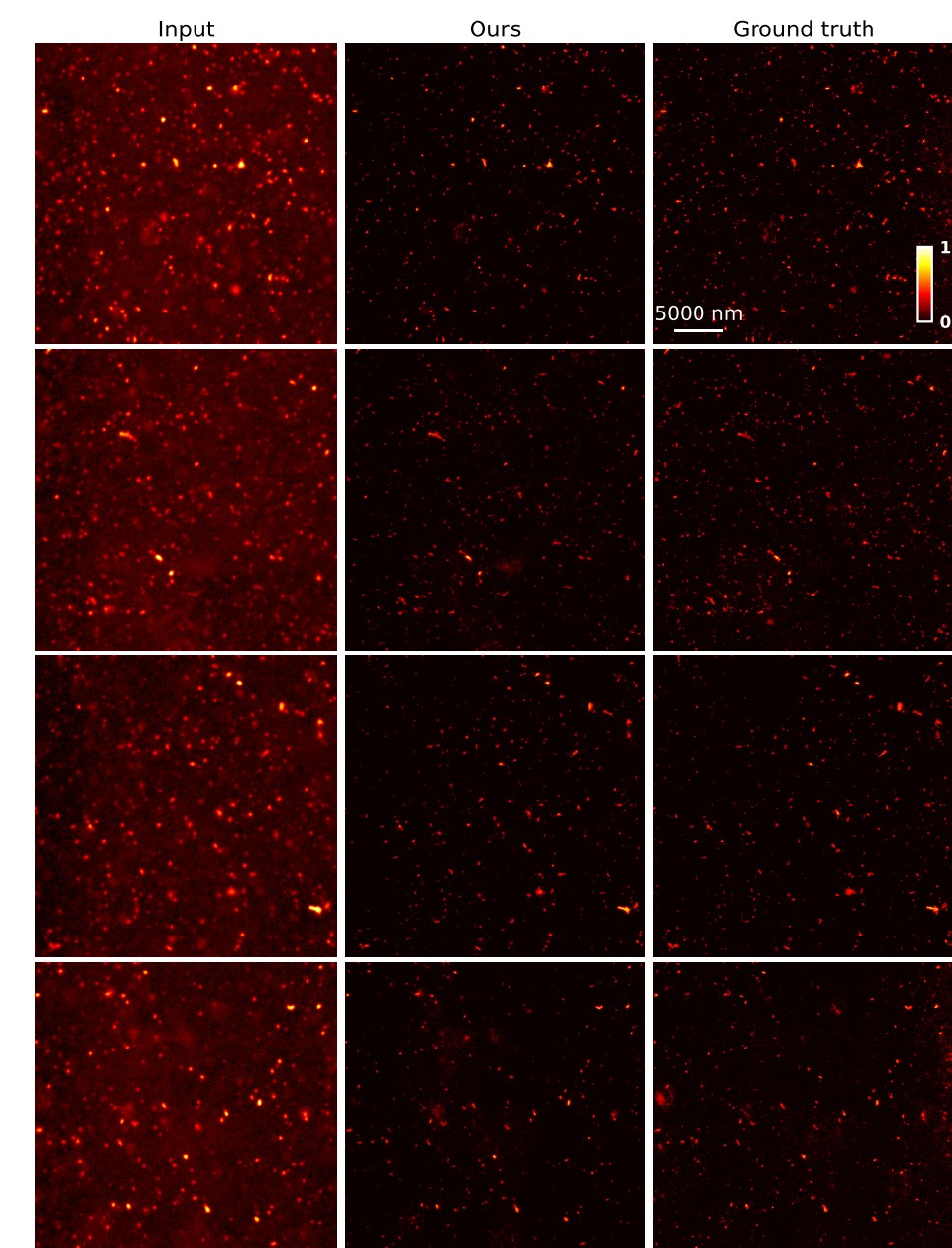

Figure 11: Additional microscopy image super-resolution results of nanobeads.

Figure 4 and 10 presents additional results on microscopy image super-resolution of HeLa cells and Fig. 11 showcases additional results on microscopy image super-resolution of nanobeads. In correspondence to the results shown in the main text, SR images generated by our method match with the ground truths very well.

### G.4 FAST SAMPLING RESULT

Here we present the sampling results of our method(MSCGM) and BBDM of each sampling steps (4, 16, 64, 256) on the Set14 and DIV2k validation dataset. Figure 12 show the sampling results of our method and BBDM with various sampling steps on Set14 dataset, and Fig. 13 illustrates the improvement of sampling quality with respect to the number of sampling steps from 4 to 1000 on

DIV2K validation dataset. Remarkbly, our method recovers high-quality image considerably faster in fewer sampling steps, confirming its superiority in sampling speed compared to competitive diffusion models.

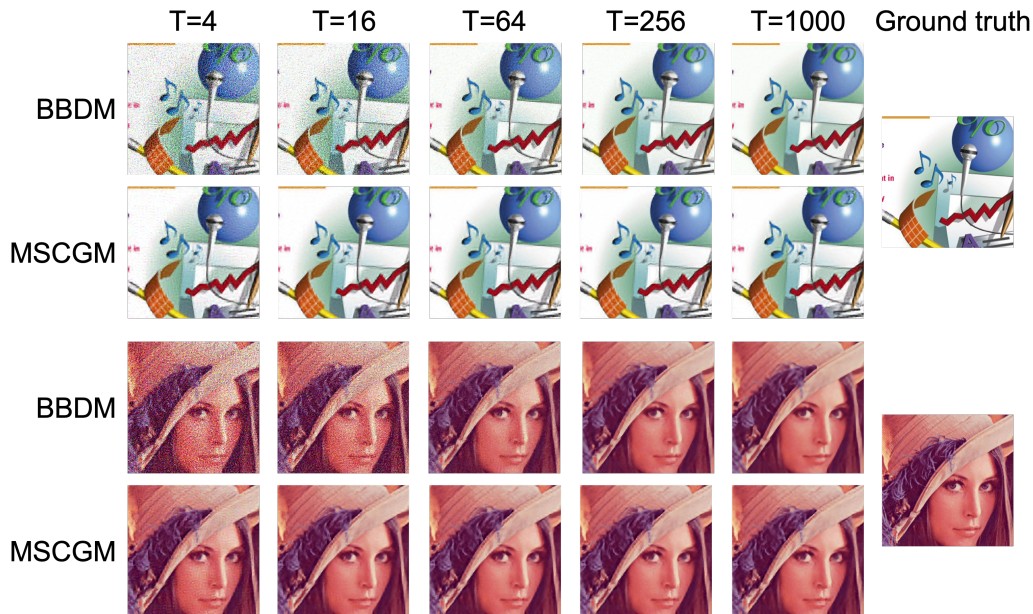

Figure 12: Sampling results of our method (MSCGM) and BBDM with various sampling steps from 4 to 1000. Images sampled from $64 \times 64$ LR images in Set14.

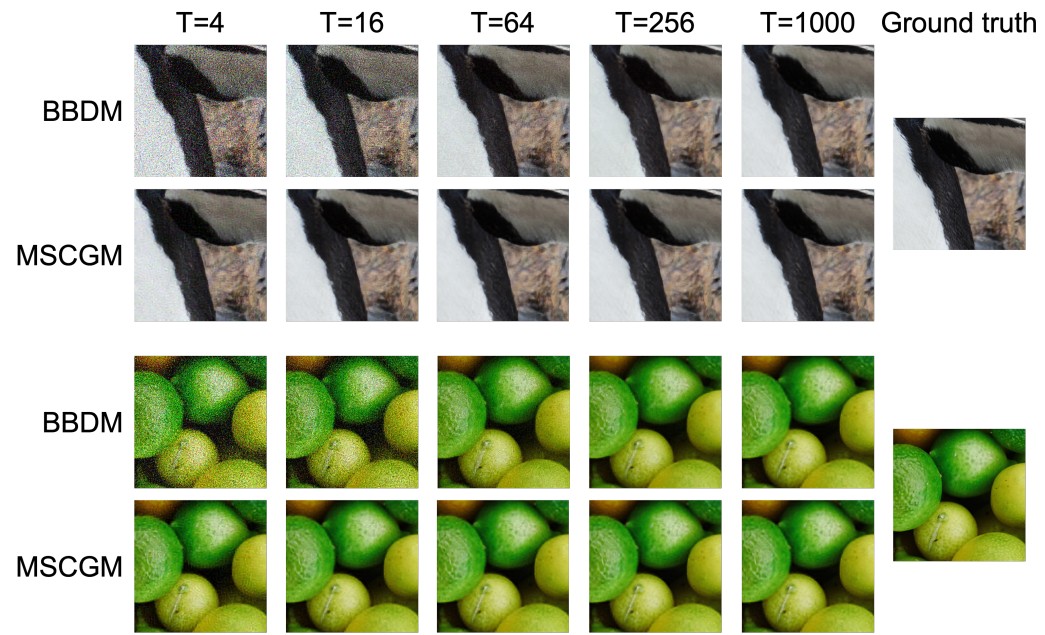

Figure 13: Sampling results of our method (MSCGM) and BBDM with various sampling steps from 4 to 1000. Images sampled from $64 \times 64$ LR images in DIV2K validation dataset.

