# OpenReview forum: "Multi-scale Conditional Generative Modeling for Microscopic Image Restoration"
_ICLR.cc/2025/Conference — Submitted to ICLR 2025_

### Official Review · Reviewer_CZbw · 2024-10-28

**Soundness:** 3
**Presentation:** 3
**Contribution:** 3
**Rating:** 6
**Confidence:** 4

**Summary:**

The paper proposed a multi-scaled generative model that uses a diffusion model (DM) for low-frequency image and a GAN for high frequency images. The wavelet transform provides multi-scale images without lossy encoding process. The lossless compression is particularly important for microscopic imaging where high-frequency component are sparse and non-Gaussian. Additionally, the authors showed the near-Gaussian property of low-frequency component and thus employed Brownian Bridge Diffusion Process (BBDP). The idea of employing different networks (DM and GAN) to different resolutions according to the characteristics of microscopic dataset is novel. The proposed MSCGM (multi-scale conditional generative model) showed improved super-resolution result with fast inference time.

**Strengths:**

The paper analyzed the characteristics of microscopic images and proposed adequate methodology to address the sparsity and non-Gaussianity. Since the wavelet transformation divides the image into two subbands (high- and low- frequency coefficients) losslessly, handling each subband in a different manner is original.
The contribution of the work is clear and well demonstrated.
In addition, the work could be further applied to different modality images where sparse or non-Gaussianity exist.

**Weaknesses:**

Although the idea of the paper is novel, the effectiveness of the work has not been thoroughly assessed. The use of WT and the superiority of the proposed method compared to conventional method should be further evaluated. The specific comments are described in Questions.

**Questions:**

The paper demonstrated that the low-frequency coefficients in higher scales show Gaussian tendency and thus applied this to BBDP. The idea is novel and well hypothesized, but it would be helpful if other DM methods, such as IR-SDE and ReFusion methods that are implemented on 4x super-resolution experiment, are also tested on microscopy image dataset. Only CMSR (GAN: non-diffusion model), is compared at the moment, not showing the effectiveness of proposed near-Gaussianity assumption.
Similarly, applying BBDM to full resolution image does not seem to be fair comparison. Since many works demonstrated the effectiveness of multi-scale diffusion models, BBDM should be implemented in a same manner as the proposed method to prove the superiority of WT instead of other compression technique. Please conduct an ablation study that replaces WT with simple down-sampling.
Is there any specific reason why the proposed work adopted BBDM which was initially designed for image translation where input and target domains are different? Super-resolution tasks seem to have similar domains for input and target. Justify the choice of BBDM for super-resolution.

---

> ### Author Response · Authors · 2024-11-28
> **Thank you for your thoughtful and constructive feedback.**
>
> We thank the reviewer for this valuable point.
>
> In microscopy image super-resolution, for example, Figure 3 and 4, the input and target images are actually captured by two different imaging modalities and belong to two distinct image domains. Therefore a structure like BBDP instead of DDPM is essential to establish mappings between the two domains. The super-resolution experiments on natural images are presented to show the versatility of our method and provide comparison against existing baselines.
>
>
> Here we provide a ablation study that replaces WT with a simple down-sampling method on Div2K dataset:
> | Methods  | PSNR (dB) ↑ (DIV2K) | PSNR (dB) ↑ (Set5) | PSNR (dB) ↑ (Set14) | SSIM ↑ (DIV2K) | SSIM ↑ (Set5) | SSIM ↑ (Set14) |
> |----------|---------------------|---------------------|-------------------|--------------------|----------------|---------------|
> |  MSCGM (simple down-sampling method)  | 30.73| 31.28        | 29.67       | 0.65       | 0.78      | 0.65      |
> | MSCGM    | **31.66**          | **32.33**        | **30.79**         | **0.72**       | **0.85**      | **0.71**      |
>
> The inferior performance of MSCGM with simple down-sampling is expected as the high-frequency signals are lost during down-sampling. And it makes GAN much harder to reconstruct a high-quality image with rich details.
>
> Thank you again for your hard work. If you have any additional questions or concerns, please don’t hesitate to reach out, we will answer them in time!

---

> ### Comment · Reviewer_CZbw · 2024-11-29
>
> Thank you for addressing the questions I raised.
> I understand the use of BBDP instead of DDPM because a different imaging technique was used for the input and target images.
> As the author mentioned, simple down-sampling drops quite a lot of information. Therefore, pixel-shuffling, as applied in “MSSNet: Multi-Scale-Stage Network for Single Image Deblurring,” should be used for a fair comparison.
> The table shows that the MSCGM is better with simple down-sampling against WT. Is it correct?

---

> > ### Author Response · Authors · 2024-11-29
> > **Thank you for your timely response!**
> >
> > Thank you very much for your timely response. We are really glad that our answer has addressed your concerns.
> >
> > We realize that we mistakenly included results from different experiments in our comparison. We have now corrected this error and updated the results accordingly.
> >
> > Once again, thank you for your correction and for your valuable effort. Your insights are greatly appreciated!

---

### Official Review · Reviewer_yXmP · 2024-11-01

**Soundness:** 1
**Presentation:** 2
**Contribution:** 2
**Rating:** 5
**Confidence:** 4

**Summary:**

The authors present a novel multi-scale generative model that leverages the Brownian Bridge process within the wavelet domain. This approach enhances inference speed while maintaining high image quality and diversity. The model is further integrated with computational microscopy workflows, expanding its applicability. The authors evaluate its performance on both microscopy and natural image datasets, demonstrating that it achieves slightly better results compared to existing methods such as IR_SDE, Refusion, and BBDM, with the added advantage of faster inference.

**Strengths:**

• Integrating the Brownian Bridge Diffusion Process (BBDP) with adversarial learning in a multi-scale wavelet diffusion framework is innovative, enhancing image quality and stability.

• The model achieves notable speed improvements, delivering faster inference without sacrificing image quality.

• Performance remains consistent across diverse experiments, demonstrating robustness on both microscopy and natural images.

**Weaknesses:**

• The paper lacks a clear motivation for applying this model to computational microscopy workflows. The rationale for this specific application is unclear and lacks context, the relevance to microscopy appears out of place. A discussion on how this functionality benefits microscopy would help justify this direction and clarify its practical utility.

• The primary advantage of this method is its reduced inference time; however, the paper lacks a direct comparison with other methods that similarly aim to improve efficiency. Including such a comparison would provide valuable context and help quantify the benefits more clearly.

• The general evaluation lacks depth and is missing ablation studies.

• There appear to be configuration issues with the comparison methods. For instance, IR-SDE [1] is cited as requiring 100 steps, but the authors use 1000, which significantly prolongs inference time. With the correct configuration (100 steps), the inference time should drop from 32 seconds to approximately 3 seconds.

• The choice of metrics is limited and somewhat inadequate for a super-resolution task. Relying solely on PSNR and SSIM may overlook important aspects of image quality. Including pixel-based metrics would provide a more comprehensive evaluation and might show shortcomings of the proposed method.


[1] Luo, Ziwei, et al. "Image restoration with mean-reverting stochastic differential equations." arXiv preprint arXiv:2301.11699 (2023).

**Questions:**

Especially considering that inference time is one of the main benefits, why was it not compared to models with fewer step counts or at least an in-depth analysis of how step counts influence the SOTA model performance? E.g. [2], or other methods that can be applied to the problem domain?

[2] Phung, Hao, Quan Dao, and Anh Tran. "Wavelet diffusion models are fast and scalable image generators." Proceedings of the IEEE/CVF conference on computer vision and pattern recognition. 2023.

---

> ### Author Response · Authors · 2024-11-28
> **Thank you for your thoughtful and constructive feedback.**
>
> We sincerely thank you for your careful review and insightful comments. We have provided a point-to-point response to each of your concerns below:
>
> >**Motivation:** The paper lacks a clear motivation for applying this model to computational microscopy workflows. The rationale for this specific application is unclear and lacks context, the relevance to microscopy appears out of place. A discussion on how this functionality benefits microscopy would help justify this direction and clarify its practical utility.
>
> **A:** Generative models such as GANs have been successfully and widely used in microscopy imaging applications. It is important to integrate the existing state-of-art diffusion models and GANs on these datasets and advance AI-enabled microscopy imaging.
>
> >**Direct comparison:** The primary advantage of this method is its reduced inference time; however, the paper lacks a direct comparison with other methods that similarly aim to improve efficiency. Including such a comparison would provide valuable context and help quantify the benefits more clearly.
>
> **A:** Our work proposes an innovative multi-scale conditional generative model for conditional image generation and translation tasks, and validates it on multiple datasets, including natural images and microscopic images. We mainly demonstrate a fundamental implementation, which is similar to DDPM, and existing acceleration techniques, including DDIM and pseudo numerical methods can be directly applied to the sampling process to improve its efficiency.
>
> >**Ablation studies:** The general evaluation lacks depth and is missing ablation studies.
>
> **A:** Thank you for your suggestion. We have added the ablation study results of the experiment mentioned in this article. We remove the wavelet transform in the model and only conduct the experiment in the time domain.
>
> | Methods  | PSNR (dB) ↑ (DIV2K) | PSNR (dB) ↑ (Set5) | PSNR (dB) ↑ (Set14) | SSIM ↑ (DIV2K) | SSIM ↑ (Set5) | SSIM ↑ (Set14) |
> |----------|---------------------|---------------------|-------------------|--------------------|----------------|---------------|
> |  MSCGM (without wavelet)  | 30.13          | 31.01       | 28.99         | 0.68      | 0.76 | 0.63|
> | MSCGM    | **31.66**          | **32.33**        | **30.79**         | **0.72**       | **0.85**      | **0.71**      |
>
>
>
> >**Configure issue:** There appear to be configuration issues with the comparison methods. For instance, IR-SDE [1] is cited as requiring 100 steps, but the authors use 1000, which significantly prolongs inference time. With the correct configuration (100 steps), the inference time should drop from 32 seconds to approximately 3 seconds.
>
> **A:** Sample steps of 1000 were employed for fair comparison. As discussed above, the same acceleration strategy, including a smaller total sample steps can be equivalently applied during the training and/or sampling of our method to reach the same reduction of time.
>
> >**Metrics:** The choice of metrics is limited and somewhat inadequate for a super-resolution task. Relying solely on PSNR and SSIM may overlook important aspects of image quality. Including pixel-based metrics would provide a more comprehensive evaluation and might show shortcomings of the proposed method.
>
> **A:** The paper referred here by Luo et al [1]. applies the same metrics PSNR and SSIM for the super-resolution task, as reported in Table 1 in our paper. We added the information about the number of model parameters in the revision as further supplement.
>
> >**Compare with different steps:** Especially considering that inference time is one of the main benefits, why was it not compared to models with fewer step counts or at least an in-depth analysis of how step counts influence the SOTA model performance? E.g. [2], or other methods that can be applied to the problem domain?
>
> **A:** Thank you for the question, Figure 12 and 14 in appendix G.4 shows the sampling results of our method (MSCGM) and BBDM with various sampling steps from 4 to 1000. This intuitively explains the performance of the model under different sample steps. Since the underlying logic of our implementation is different from that of wavelet diffusion [2], we cannot simply apply a different number of steps for comparison, which is relatively unfair.
>
> Thank you again for your valuable feedback and suggestions. If you have any additional questions or concerns, please don’t hesitate to reach out. If our responses have addressed most of your concerns, we kindly ask if it might improve your evaluation of our manuscript. Your support is greatly appreciated!
>
>
> *[1] Luo, Ziwei, et al. "Image restoration with mean-reverting stochastic differential equations." arXiv preprint arXiv:2301.11699 (2023).*
>
> *[2] Phung, Hao, Quan Dao, and Anh Tran. "Wavelet diffusion models are fast and scalable image generators." Proceedings of the IEEE/CVF conference on computer vision and pattern recognition. 2023.*

---

### Official Review · Reviewer_Ytoo · 2024-11-02

**Soundness:** 2
**Presentation:** 1
**Contribution:** 2
**Rating:** 3
**Confidence:** 5

**Summary:**

The authors propose a multi-scale conditional generative model (MSCGM) for image restoration, incorporating multi-scale wavelet transforms and a Brownian bridge stochastic process. The wavelet transform is included due to its reversibility, which maintains information integrity in the latent diffusion space, in contrast to traditional Latent Diffusion Models (LDM). The Brownian bridge stochastic process is leveraged to introduce conditional images in both forward and reverse processes. While the authors aim to address microscopic image restoration, the motivation and results in the paper do not consistently support this focus.

**Strengths:**

1. The authors recognize the loss of detail in LDM, a known issue, and apply it to the microscopic image restoration context, an interesting direction.
2. They introduce the novel idea that the Brownian bridge stochastic process could effectively integrate conditional images.

**Weaknesses:**

1. **Lack of Consistency:** The paper lacks organization and clarity. Although the title emphasizes "Microscopic Image Restoration," the experiments primarily focus on "Natural Image Super-resolution" and "Low-light Natural Image Enhancement." Only a small subset of results explores microscopic images. If the model is intended for general image restoration, it would be more accurate to propose it as a ‘unified image restoration’ model. I suggest the authors either refocus their experiments more heavily on microscopic image restoration to align with the title, or broaden the title to reflect the wider scope of image restoration tasks covered in the paper.

2. **Introduction Needs Refinement:** The introduction lacks a clear problem definition and research motivation. The first two paragraphs provide a broad overview of diffusion processes that diverges from the paper’s focus. The discussion on latent diffusion downsampling is a well-known issue and could be alleviated by higher resolutions. The authors should clearly articulate why microscopic images especially require the multi-scale wavelet transform in the introduction. Please include a discussion of how their approach compares to or builds upon these existing wavelet-based diffusion models in the Introduction, highlighting any key differences or improvements.

3. **Lack of Acknowledgment of Prior Work:** The paper does not credit previous studies applying wavelet transforms in diffusion models, which could mislead readers into believing the concept originated here. Papers like "Wavelet Diffusion Models are Fast and Scalable Image Generators (CVPR 2023)" and "Training Generative Image Super-Resolution Models by Wavelet-Domain Losses Enables Better Control of Artifacts (CVPR 2024)" are directly related and should be cited with comparisons to clarify this study’s contributions.

4. **Figure 1 Illustration Issues:** The paper title focuses on "Microscopic Image Restoration," yet Figure 1 uses natural images. Including examples of microscopic images to show the degradations introduced by LDM and Refusion compared to MSCGM would enhance clarity.

5. **Methodology Development Clarity:** The description of the wavelet transform on page 4 is overly general, with key details moved to the appendix. Clear explanations of any novel model designs or algorithmic adaptations should be provided in the main text.

6. **Quality of Mathematical Presentation:** Symbols in the equations are used without proper declarations or explanations. Inconsistent symbols, like the variable for the normal distribution \( N \), further detract from clarity.

7. **Algorithm 1 Lack of Context:** Algorithm 1 on page 5 is underdeveloped. Symbols are not defined before use, and the algorithm lacks defined input-output requirements.

8. **Figure 2 Diagram Confusion:** Figure 2 is difficult to interpret. The illustration doesn’t clearly label network modules, workflow processes, or shared parameters (only a line is shown), which fails to clarify the model structure effectively.

9. **Lack of Dataset Information:** The results section includes evaluations of microscopic images, but there’s no description of the dataset. Is it public or private? What is the image count? Without these details, readers cannot analyze or reproduce the results. Please provide a detailed description of the microscopic image dataset used, including its source, size, and any preprocessing steps applied.

10. **Insufficient Ablation Studies:** Results provide only a simple comparison with LDM, without deeper exploration of MSCGM’s components or ablation studies to justify the performance benefits of each module.

11. **Unconvincing Model Performance:** The model’s performance requires further validation through comparison with advanced models. Numerous diffusion-based image restoration models from 2024 exist, yet none are used for comparison. This weakens the paper’s credibility. Key diffusion-based image restoration works worth considering include:
   - RDDM ([link](https://cvpr.thecvf.com/virtual/2024/poster/31373))
   - HIR-Diff ([link](https://cvpr.thecvf.com/virtual/2024/poster/29665))
   - WF-Diff ([link](https://cvpr.thecvf.com/virtual/2024/poster/30059))
   - DeqIR ([link](https://cvpr.thecvf.com/virtual/2024/poster/31759))
   - GDP ([link](https://cvpr.thecvf.com/virtual/2023/poster/22095))

**Questions:**

Please see my concerns in Weakness

---

> ### Author Response · Authors · 2024-11-28
> **Thank you for your through review and encouraging feedback.**
>
> We thank you for your thorough review and constructive comments on our work. In response to your questions/concerns, we have provided detailed answers below.
>
> >**Lack of Consistency:** The paper lacks organization and clarity. Although the title emphasizes "Microscopic Image Restoration," the experiments primarily focus on "Natural Image Super-resolution" and "Low-light Natural Image Enhancement." Only a small subset of results explores microscopic images. If the model is intended for general image restoration, it would be more accurate to propose it as a ‘unified image restoration’ model. I suggest the authors either refocus their experiments more heavily on microscopic image restoration to align with the title, or broaden the title to reflect the wider scope of image restoration tasks covered in the paper.
>
> **A:** Our theory and model are mainly aimed at microscopic images with more details but sparse image contents. The applications on natural image datasets are shown for two reasons: (1) on these tasks we have pre-trained baselines to establish solid a comparison between our method and previous state-of-the-art methods, (2) to demonstrate the adaptability of our method on non-sparse, general image restoration tasks such as natural image super-resolution.
>
> Besides, the two microscopic image datasets involved in this work are representative as they (1) are captured by a typical super-resolution optical microscopy (stimulated emission depletion microscopy) for the HR images and a diffraction-limited microscopy (confocal) for the LR images, (2) contain typical sparse samples with distinct features, including simple samples like nano-beads and relatively complex samples like HeLa cells.  The main focus of this article is still on the super-resolution task of microscopic images.
>
> >**Introduction Needs Refinement:** The introduction lacks a clear problem definition and research motivation. The first two paragraphs provide a broad overview of diffusion processes that diverges from the paper’s focus. The discussion on latent diffusion downsampling is a well-known issue and could be alleviated by higher resolutions. The authors should clearly articulate why microscopic images especially require the multi-scale wavelet transform in the introduction. Please include a discussion of how their approach compares to or builds upon these existing wavelet-based diffusion models in the Introduction, highlighting any key differences or improvements.
>
> **A:** Our introduction strictly focuses on the microscopic image restoration tasks, discusses and compares existing methods, including GAN and diffusion models, the motivations of this work, especially why we design MSCGM specifically for microscopy data, is elucidated in the second to last paragraph of the Introduction section, cited below:
> “On the other hand, …”
> The advantages of our method over existing ones are summarized in the last paragraph of the Introduction section, cited below:
> “”
> To the best of our knowledge, related methods like wavelet-based DMs were not designed for or demonstrated on microscopy data, therefore we believe such a comparison or claim in the Introduction is improper. Detailed discussion on related works such as wavelet-based DM are elucidated in the “Related Works” section.
>
>
> >**Lack of Acknowledgement of Prior Work:** The paper does not credit previous studies applying wavelet transforms in diffusion models, which could mislead readers into believing the concept originated here. Papers like "Wavelet Diffusion Models are Fast and Scalable Image Generators (CVPR 2023)" and "Training Generative Image Super-Resolution Models by Wavelet-Domain Losses Enables Better Control of Artifacts (CVPR 2024)" are directly related and should be cited with comparisons to clarify this study’s contributions.
>
> **A:** Thank you for mentioning these two previous related works. We have already cited the first work in line 164 that you have mentioned and added the second work in the related work part of our revision. Thank you again for your suggestions, which help improve the completeness of our paper.

---

> ### Author Response · Authors · 2024-11-28
> **Cont.**
>
> >**Figure 1 Illustration Issues:**  The paper title focuses on "Microscopic Image Restoration," yet Figure 1 uses natural images. Including examples of microscopic images to show the degradations introduced by LDM and Refusion compared to MSCGM would enhance clarity.
>
> **A:** Thank you for your insightful comment regarding Figure 1. We chose to use natural images in this figure because they effectively demonstrate the significant differences between wavelet transform and the other two methods. While using microscopic images is feasible, they do not clearly illustrate the lossless property of the wavelet transform. Moreover, our intention with Figure 1 was just to provide a simple illustration of the pattern. (LDM and Refusion were not designed for microscopy data and no public implementation was available) We appreciate your suggestion and will consider it in future presentations to enhance clarity.
>
> >**Methodology Development Clarity:** The description of the wavelet transform on page 4 is overly general, with key details moved to the appendix. Clear explanations of any novel model designs or algorithmic adaptations should be provided in the main text.
>
> **A:** Thank you for your suggestion. We will move the novel model designs or algorithmic adaptations in the appendix to the main text.
>
> >**Quality of Mathematical Presentation:** Symbols in the equations are used without proper declarations or explanations. Inconsistent symbols, like the variable for the normal distribution ( N ), further detract from clarity.
>
> **A:** Thank you for pointing out this problem. We have checked the consistency of the equations in the revision and corrected the inconsistencies.
>
> >**Algorithm 1 Lack of Context:** Algorithm 1 on page 5 is underdeveloped. Symbols are not defined before use, and the algorithm lacks defined input-output requirements.
>
> **A:** We added more explanatory text.
>
> >**Figure 2 Diagram Confusion:** Figure 2 is difficult to interpret. The illustration doesn’t clearly label network modules, workflow processes, or shared parameters (only a line is shown), which fails to clarify the model structure effectively.
>
> **A:** Modify Fig.2 to add a GAN module. It is common practice to omit the tedious model structures but elucidate them in the Methods section.

---

> ### Author Response · Authors · 2024-11-28
> **Cont.**
>
> >**Lack of Dataset Information:** The results section includes evaluations of microscopic images, but there’s no description of the dataset. Is it public or private? What is the image count? Without these details, readers cannot analyze or reproduce the results. Please provide a detailed description of the microscopic image dataset used, including its source, size, and any preprocessing steps applied.
>
> **A:** Thank you for your concern. However, a detailed description of our dataset and parts of the dataset are already provided in Appendix I. You can check the link we provided.
>
> >**Insufficient Ablation Studies:** Results provide only a simple comparison with LDM, without deeper exploration of MSCGM’s components or ablation studies to justify the performance benefits of each module.
>
> **A:** Thank you for your suggestion. We have added the ablation study results of the experiment mentioned in this article. We remove the wavelet transform in the model and only conduct the experiment in the time domain and use the different down-sampling method. Results are provided as below:
>
> | Methods  | PSNR (dB) ↑ (DIV2K) | PSNR (dB) ↑ (Set5) | PSNR (dB) ↑ (Set14) | SSIM ↑ (DIV2K) | SSIM ↑ (Set5) | SSIM ↑ (Set14) |
> |----------|---------------------|---------------------|-------------------|--------------------|----------------|---------------|
> |  MSCGM (without wavelet)  | 30.13          | 31.01       | 28.99         | 0.68      | 0.76| 0.63|
> |  MSCGM (simple down-sampling method)  | 30.73| 31.28        | 29.67       | 0.70      | 0.78      | 0.65      |
> | MSCGM    | **31.66**          | **32.33**        | **30.79**         | **0.72**       | **0.85**      | **0.71**      |
>
> >**Unconvincing Model Performance:** The model’s performance requires further validation through comparison with advanced models. Numerous diffusion-based image restoration models from 2024 exist, yet none are used for comparison. This weakens the paper’s credibility. Key diffusion-based image restoration works worth considering include:
>
> **A:** Due to the limitation of running resources, we may not be able to verify all the proposed models one by one, but our experiments show that we perform best on this special type of microscopic images with high details and sparseness. Our experiments mainly provide support and verification for the theory of multiscale conditional generative modeling, which is a basic model that can be applied to more advanced model structures. In addition, for more experiments details, please refer to Appendix G.3 and the result table in the article. At the same time, the outstanding advantage of our model is fast sampling. In this regard, our model performs much better than other models.
>
> We hope our answers can answer your doubts and concerns. If you have further questions, please do not hesitate to ask us. We will answer all your questions in a timely manner. Thanks again for your hard work!

---

### Official Review · Reviewer_a2Tp · 2024-11-04

**Soundness:** 3
**Presentation:** 3
**Contribution:** 2
**Rating:** 6
**Confidence:** 5

**Summary:**

The paper introduces a multi-scale conditional generative model (MSCGM) aimed at enhancing microscopic image restoration by combining wavelet transforms and a Brownian Bridge diffusion process. The authors leverage multi-scale wavelet transforms to efficiently model low- and high-frequency image components, significantly improving the generation quality and speed of image restoration compared to traditional diffusion models.

**Strengths:**

1. MSCGM’s wavelet-based decomposition and conditional modeling shows substantial improvements in sampling speed and better reconstruction quality.
2. By adapting the generative approach to frequency characteristics, MSCGM enhances detail in restored images, especially in high-frequency components crucial for microscopy images.
3. The authors presented a new loss function.

**Weaknesses:**

1. Equation 18 combines multiple objectives—L2 loss, Structural Similarity Index Measure (SSIM), and Wasserstein distance—but the rationale behind each component’s inclusion is not fully explained. Additionally, the roles and relative importance of the scaling parameters λ, ν, and α are unclear.
2. The training procedure for MSCGM is not explicitly described. Unlike the clear training steps outlined for BBDP, MSCGM lacks a step-by-step description of its training pipeline.
3. While Table 1 compares MSCGM with other models in terms of PSNR, SSIM, and sampling time, it does not include training time or the number of trainable parameters for each method. Without these metrics, it is challenging to gauge MSCGM’s overall computational cost relative to other approaches. Including such details would provide a more comprehensive view of the model’s efficiency.
4. In Section 4.2, the authors state that FID is considered as an evaluation metric. However, this metric is not included in Table 1. As FID is widely used in assessing generative models for image quality, its inclusion would offer further insights into MSCGM’s performance in distributional similarity to real images.
5. Equations from 4 to 15 are borrowed from BBDP paper. It is better to include them under the Preliminaries section.

**Questions:**

1. Could the authors provide more detailed explanations regarding the choice and role of each loss term in Equation 18 and explain how they determined the relative weighting (λ, ν, α values) between the terms.
2. Could the authors provide a comparison of training time and the number of training parameters for MSCGM versus other models?
3. Could the authors to provide a detailed algorithm or pseudocode for MSCGM training, similar to what they provided for BBDP Algorithm.

**Details Of Ethics Concerns:**

The anonymity of the authors is compromised, as this paper is available on arXiv at https://arxiv.org/abs/2407.05259.

---

> ### Author Response · Authors · 2024-11-28
> **Thank you for your through review and encouraging feedback.**
>
> We would like to thank you for acknowledging the novelty and the significance of our work. We now take the opportunity to clarify the raising concerns:
> >**Training loss explanation:** Could the authors provide more detailed explanations regarding the choice and role of each loss term in Equation 18 and explain how they determined the relative weighting (λ, ν, α values) between the terms.
>
>
> **A:** These loss terms have been widely used in previous works, such as:
>
>
> 1. [Zhang H, et al. High-throughput, high-resolution deep learning microscopy based on registration-free generative adversarial network. Biomed Opt Express. 2019 Feb 4;10(3):1044-1063. doi: 10.1364/BOE.10.001044.]
>
>
> 2. [Yilin Luo, et al.Single-Shot Autofocusing of Microscopy Images Using Deep Learning ACS Photonics 2021 8 (2), 625-638 DOI: 10.1021/acsphotonics.0c01774]
>
>
> 3. [Xu, Hao, et al. "Microscopic image augmentation using an enhanced WGAN." The fourth international symposium on image computing and digital medicine. 2020.]
>
>
> And parameters vary for each experiment and are generally determined empirically. To better clarify the functions of each loss term, we have revised the Methods section and add a sentence above the loss definition, cited below:
>
>
> *“We adopted a pixel-wise L2 loss and a structural similarity index loss to penalize local and global mismatch, respectively.”*
>
>
> >**Training procedure description:** The training procedure for MSCGM is not explicitly described. Unlike the clear training steps outlined for BBDP, MSCGM lacks a step-by-step description of its training pipeline.
>
>
> **A:** We thank the reviewer for this valuable suggestion. We have added the detailed description of the training procedure of MSCGM in the appendix.
>
>
> >**Adding more metrics:** Could the authors provide a comparison of training time and the number of training parameters for MSCGM versus other models?
>
>
> **A:** We utilized publicly available implementations of IR-SDE and ReFusion. The training time was not provided. But we have revised Table 1 to include the comparison on trainable parameters for the four models.
>
>
>
>
> >**FID:** In Section 4.2, the authors state that FID is considered as an evaluation metric. However, this metric is not included in Table 1. As FID is widely used in assessing generative models for image quality, its inclusion would offer further insights into MSCGM’s performance in distributional similarity to real images.
>
>
> **A:** For image restoration tasks such as image super-resolution tasks, PSNR and SSIM are more commonly used metrics to evaluate the similarity between outputs and ground truth images. In contrast, FID focuses on measuring the distribution similarity between output and target images. We also report FID metrics for some tasks in the appendix.
>
>
> >**Reintegrate content:** Equations from 4 to 15 are borrowed from BBDP paper. It is better to include them under the Preliminaries section.
>
>
> **A:** We really thank the suggestions from the reviewer. However, we found merging section 3.1 and 3.2 makes the preliminaries too prolonged. In order to explain the content more clearly, we decided to separate them.
>
>
> >**Detailed algorithm:** Could the authors to provide a detailed algorithm or pseudocode for MSCGM training, similar to what they provided for BBDP Algorithm.
>
>
> **A:** Thank you for the heads up, we have included the pseudocode in the appendix.
>
>
>
>
> We hope our answers can answer your doubts and concerns. If you have further questions, please do not hesitate to ask us. We will answer all your questions in a timely manner. Thanks again for your hard work!

---

### Meta-Review · Area_Chair_HFR9 · 2024-12-22

**Metareview:**

This work introduces a multi-scale generative model to enhance conditional image restoration by initiating the Brownian Bridge diffusion process specifically at the lowest-frequency subband and applying generative adversarial networks at subsequent multi-scale high-frequency subbands in the wavelet domain.

**Additional Comments On Reviewer Discussion:**

This work has four reviewers. Two reviewers are positive to accept it, while the other two reviewers are negative to accept it. And the final ratings after rebuttal are 6, 6, 3, and 5. After checking the comments and the author's responses, I find that this work has many weaknesses about unclear motivations, clarification issues, unconvinced experiments without comparisons against SOTA diffusion-based methods, missed ablation studies, and so on. In this regard, this work can not be accepted in ICLR 2015.

---

### Decision · Program_Chairs · 2025-01-22

Reject